# Imitation Learning from Vague Feedback

**Xin-Qiang Cai[1], Yu-Jie Zhang[1], Chao-Kai Chiang[1], Masashi Sugiyama[2,1]**
[1] The University of Tokyo, Tokyo, Japan
[2] RIKEN AIP, Tokyo, Japan

## Abstract

Imitation learning from human feedback studies how to train well-performed imitation agents with an annotator's relative comparison of two demonstrations (one demonstration is better/worse than the other), which is usually easier to collect than the perfect expert data required by traditional imitation learning. However, in many real-world applications, it is still expensive or even impossible to provide a clear pairwise comparison between two demonstrations with similar quality. This motivates us to study the problem of imitation learning with vague feedback, where the data annotator can only distinguish the paired demonstrations correctly when their quality differs significantly, i.e., one from the expert and another from the non-expert. By modeling the underlying demonstration pool as a mixture of expert and non-expert data, we show that the expert policy distribution can be recovered when the proportion $\alpha$ of expert data is known. We also propose a mixture proportion estimation method for the unknown $\alpha$ case. Then, we integrate the recovered expert policy distribution with generative adversarial imitation learning to form an end-to-end algorithm[1]. Experiments show that our methods outperform standard and preference-based imitation learning methods on various tasks.

## 1 Introduction

Imitation learning (IL) is a popular approach for training agents to perform tasks by learning from expert demonstrations [1–3]. It has been applied successfully to a variety of tasks, including robot control [1], autonomous driving [4], and game playing [5]. However, traditional IL methods struggle when presented with both expert and non-expert demonstrations, as the agents may learn incorrect behaviors from the non-expert demonstrations [6]. This problem, which researchers refer to as "Imitation Learning from Imperfect Demonstration" (ILfID), arises when the demonstrations used to train the model may contain some non-expert data [6–9].

One popular solution to ILfID is resorting to an oracle to provide specific information, such as explicit labels (confidence from an expert) of each demonstration, as in Figure 1a. However, such a specific oracle is quite expensive. A more recent framework, Reinforcement Learning from Human Feedback (RLHF) [10, 11], incorporates human feedback into the learning process, including a key part of the pipeline used for training ChatGPT [12]. There exist two widely studied paradigms of feedback: full ranking and global binary comparison, as in Figures 1b and 1c. Methods that use full-ranked demonstrations assumed that the feedback between every pair of trajectories is available, and further employed preference-based methods to solve the problem [13, 14]. Other studies have investigated situations where demonstrations can be divided into two global binary datasets, allowing the learner to filter out non-expert data from these global comparisons [7, 8]. However, data processed by any feedback of Figures 1a, 1b, and 1c require the guarantee that clear ranking information or at least one set consisting of pure expert demonstrations exists, so that off-the-shall IL methods can be applied immediately. The availability of expert demonstrations raises the question of what if, in practice, the feedback is not clear enough to provide purely expert demonstrations but mixed with non-expert demonstrations? The question poses a challenge that previous methods fail to address since none of

---

[1]The code is available on `https://github.com/caixq1996/COMPILER`.

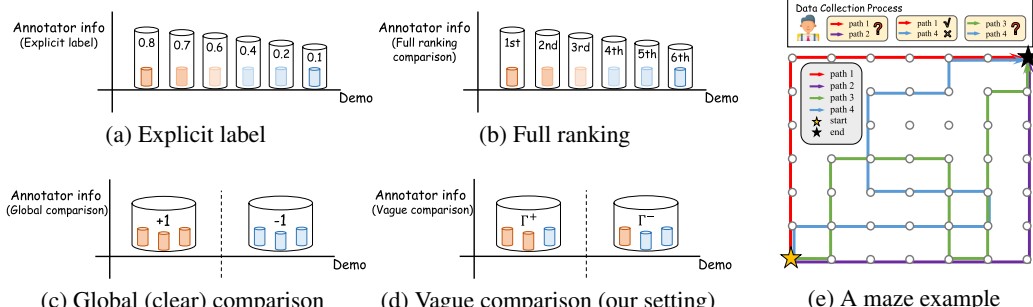

Figure 1: (a)–(d): The comparisons among the imitation learning with the explicit label, full ranking, global comparison, and our setting. 🟠 denotes the expert data while 🔵 denotes the non-expert one. (e) An example of labeling route navigation. In the example, Path 1 and Path 2 share the same distance. A similar situation lies in Path 3 and Path 4. The data collectors cannot provide explicit, ranking, or pairwise information as in (a), (b), and (c), while they can only provide vague comparisons that $\Gamma^+$ can be more expert-like than $\Gamma^-$.

the previous IL algorithms handles the issue of the *mixture* of expert and non-expert demonstrations without other information. Their conventional strategies are to deal with the *vagueness* in the data processing stage (relying on human feedback) not during the learning stage (by the learning agent).

Therefore, we consider a kind of weaker feedback than those previously formulated in the literature, which unintentionally mixes expert and non-expert demonstrations due to vague feedback. Specifically, the human annotator demonstrates in a pairwise way and only distinguishes the expert demonstration from the non-expert one. However, the annotator cannot tell the origin when both are from an expert or a non-expert. As depicted in Figure 1d, this annotation process results in two datasets: $\Gamma^+$, containing demonstrations the annotator believes to be most likely from experts, and $\Gamma^-$, containing non-expert-like demonstrations. If one only places the distinguishable demonstrations into the pools and discards the indistinguishable ones, it results in Figure 1c. Since we aim to investigate the potential of IL under the presence of non-expert demonstrations, we distribute the indistinguishable demonstrations randomly, one to $\Gamma^+$ and the other to $\Gamma^-$. This step embodies the *vagueness* we wish to study in this paper. Note that either dataset contains non-expert demonstrations, and thus, a direct application of the existing IL method might be inappropriate due to the danger of learning from undesirable demonstrations.

*Vague feedback* commonly exists in many scenarios, especially regarding RLHF [12]. In a crowd-sourcing example of routes navigation, as shown in Figure 1e, most of the time crowd-workers may lack domain expertise. Meanwhile, it is quite difficult to distinguish every pair of demonstrations (when facing two trajectories from the same source, such as (🟠,🟠) or (🔵,🔵)). Therefore, we cannot obtain high-quality crowd-sourcing labels when the data are annotated [15] as in Figures 1a, 1b, and 1c. On the other hand, it is natural for workers to provide *vague* comparisons as in Figure 1d.

In this work, we formulate the problem of *Vaguely Pairwise Imitation Learning (VPIL)*, in which the human annotator can only distinguish the paired demonstrations correctly when their quality differs significantly. In Section 4, we analyze two situations within this learning problem: VPIL with known expert ratio $\alpha$ and unknown $\alpha$. For VPIL with known $\alpha$, we provide rigorous mathematical analysis and show that the expert policy distribution can be empirically estimated with the datasets $\Gamma^+$ and $\Gamma^-$; for the more challenging situation of VPIL with unknown $\alpha$, we propose a reduction of the problem of estimating $\alpha$ to a mixture proportion estimation problem [16] and develop an iterative update procedure to estimate $\alpha$. In Section 5, we integrate our algorithm with an off-the-shelf IL method to solve VPIL problems. In Section 6, we evaluate our methods with state-of-the-art ILfID methods on a variety of tasks of MuJoCo [17] with different $\alpha$ ratios and find that our methods obtained the best performance.

## 2 Related Work

In imitation learning scenarios, a model is trained to mimic the actions of expert demonstrations. One of the main challenges of imitation learning is that the gathered demonstrations can be imprecise, making it difficult for the learner to accurately replicate the underlying policy of the demonstrations. To overcome this issue, various works have utilized an annotator, a source of supplementary supervi-

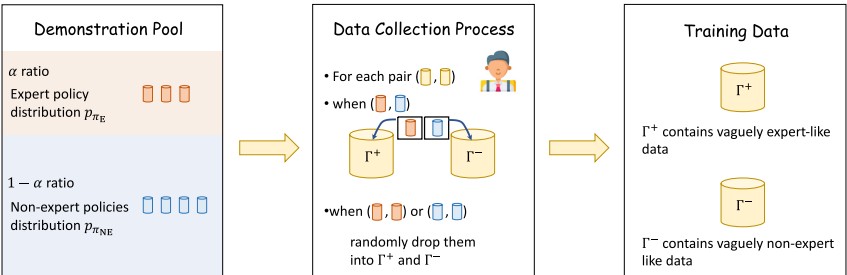

Figure 2: The description of the data collection process.

sion that can aid the learner in better understanding the expert's intent. For example, [18] used explicit action signals of the annotator; [14], [19], and [13] used ranking/preference information from the annotator to learn the policy; [6] utilized confidence information from the annotator to re-weight the unlabeled demonstrations and further learned the policy. However, as illustrated in Section 1, in some cases, the annotator may not be able to provide explicit information for the imperfect demonstrations, especially when the demonstrations come from different sources.

Alternative strategies for IL by imposing prior knowledge instead of an annotator have also been put forward, such as state density estimation for importance weighting [20], state-action extrapolating [21], and adding noise to recover ranking [22]. But this research line relies on certain assumptions about the state/state-action spaces, which may not always hold in many real-world applications. In this work, we focus on using an annotator with vague information, which is also low-cost.

Our methods drew inspiration from the risk rewriting technique in weakly supervised learning literature [23], which aims to establish an unbiased risk estimator for evaluating the model's quality only with weak supervision. Although the technique has been successfully applied to various specific weakly supervised learning problems [24–27], these methods typically require knowledge of a parameter called the mixture proportion, i.e., the expert ratio in our problem [28], the estimation of which can be challenging in the imitation learning problem (see Section 6.3 for more details). To overcome this issue, we introduce an iterative data selection procedure to better estimate the expert ratio, leading to superior empirical performance. It is worth noting that [6] also used the risk rewriting technique to identify the expert demonstrations, but their algorithm does not require estimation of the expert ratio as the confidence score is given. However, such information is unavailable in VPIL, making the problem more challenging.

# 3 Preliminaries and Problem Setting

In this section, we first introduce the IL process. Then we formulate the VPIL problem.

## 3.1 Preliminaries

**Markov Decision Process.** In policy learning problems, a Markov Decision Process (MDP) can be represented by a tuple $\langle \mathcal{S}, \mathcal{A}, \mathcal{P}, \gamma, r, T \rangle$, where $\mathcal{S}$ is the state space; $\mathcal{A}$ is the action space; $\mathcal{P} : \mathcal{S} \times \mathcal{A} \times \mathcal{S} \to [0, 1]$ is the transition probability distribution; $\gamma$ is a discount factor in the range $(0, 1]$; $r : \mathcal{S} \to \mathbb{R}$ is the reward function; and $T$ is the horizon. The goal is to learn a policy $\pi$ that maximizes the expected returns $\mathbb{E}[\sum_{t=0}^{\infty} \gamma^t r(s_t, a_t)]$, where $\mathbb{E}$ denotes the expectation.

In IL, the learner does not have access to the reward function $r$. Instead, the learner is given $m$ expert demonstrations $\tau_{\mathsf{E},1}, \tau_{\mathsf{E},2}, \ldots, \tau_{\mathsf{E},m}$, where $\tau_{\mathsf{E},i}, i \in \{1, \ldots, m\}$ is the $i$-th trajectory (a series of state-action pairs) drawn independently following the demonstrator's policy $\pi_{\mathsf{E}}$. The goal of the learner is to learn a policy $\pi_\theta$ to mimic $\pi_{\mathsf{E}}$.

**Occupancy Measure.** Since in IL we do not have reward functions, we need to measure the policy performance in the state-action space, i.e., occupancy measure $\rho_\pi$. The occupancy measure $\rho_\pi : \mathcal{S} \times \mathcal{A} \mapsto \mathbb{R}$ was introduced to characterize the distribution of state-action pairs generated by a given policy $\pi$, which was defined with the discount factor: $\rho_\pi(s, a) = \pi(a|s) \sum_{t=1}^{\infty} \gamma^t \Pr[s_t = s | \pi]$, where $\Pr[s_t = s \mid \pi]$ is the probability of reaching state $s$ at time $t$ following policy $\pi$ [29].

Moreover, we know that there is a one-to-one correspondence between the occupancy measure and the policy. Specifically, we have the following theorem.

**Theorem 1** (Theorem 2 of [29]). *Suppose $\rho$ is the occupancy measure for $\pi_\rho(a \mid s) := \frac{\rho(s,a)}{\sum_{a' \in \mathcal{A}} \rho(s,a')}$. Then, $\pi_\rho$ is the only policy whose occupancy measure is $\rho$.*

We can also define a normalized version of the occupancy measure by $p_\pi(s,a) \triangleq \frac{\rho_\pi(s,a)}{\sum_{s' \in \mathcal{S}, a' \in \mathcal{A}} \rho_\pi(s',a')} = (1-\gamma)\rho_\pi(s,a)$. If we have $m$ expert demonstrations $\tau_{\mathsf{E},1}, \tau_{\mathsf{E},2}, \ldots, \tau_{\mathsf{E},m}$, the occupancy measure $p_{\pi_\mathsf{E}}$ can be empirically estimated by the trajectories as $\widehat{p}_{\pi_\mathsf{E}}(s,a) = \frac{1-\gamma}{m} \sum_{i=1}^{m} \sum_{t=0}^{\infty} \gamma^t \mathbb{1}\left[ (s_t^{(i)}, a_t^{(i)}) = (s,a) \right]$, where $\mathbb{1}[\cdot]$ is an indicator function that returns 1 if $\cdot$ is true and returns 0 otherwise. Finding out the underlying $p_{\pi_\mathsf{E}}$ is the key for solving IL problems [1, 30].

### 3.2 Vaguely Pairwise Imitation Learning

In this work, we focus on the ILfID problem with pairwise information, where the learner aims to learn a good policy with a pairwise dataset $(\Gamma^+, \Gamma^-)$, generated from a mixture demonstration pool performed by an expert policy $\pi_\mathsf{E}$ and a set of non-expert policies $\{\pi_{\mathsf{NE}}^{(k)}\}_{k=1}^{K}$. In this work, we consider the mixed occupancy measure of the non-expert policy set as $p_{\pi_{\mathsf{NE}}}$. The proportion of the expert data within the pool is denoted as $\alpha \in (0, 1]$. The data collection process is shown in Figure 2. To clearly reveal the effect of mixed demonstrations and explain the proposed framework, we choose to assume that the data collector is not an attacker and will make mistakes, so there will be no noise during the collection process. In Section 6.4, we discuss how our method can easily be extended to handle the presence of human error.

Also, in this work, we do not consider that the data collector could be an attacker or make mistakes, so there will be no noise during the collection process.

**Collection of Pairwise Datasets.** The data collector would first sample a pair of trajectories $(\tau_i, \tau_j)$ independently from the mixture pool. If $(\tau_i, \tau_j)$ are from different sources, i.e., one is from the expert, and another is not, then the collector will take the expert one $\tau_i$ into $\Gamma^+$ and the non-expert one $\tau_j$ into $\Gamma^-$. Otherwise, the collector randomly puts them into $\Gamma^+$ and $\Gamma^-$. Under such a data generation process, the expertise probabilities of this pair are as follows:

$$\begin{cases} \Pr[\tau_i \sim p_{\pi_\mathsf{E}}, \tau_j \sim p_{\pi_\mathsf{E}}] = \alpha^2, \\ \Pr[\tau_i \sim p_{\pi_{\mathsf{NE}}}, \tau_j \sim p_{\pi_{\mathsf{NE}}}] = (1-\alpha)^2, \\ \Pr[\tau_i \sim p_{\pi_\mathsf{E}}, \tau_j \sim p_{\pi_{\mathsf{NE}}}] = 2\alpha(1-\alpha). \end{cases} \tag{1}$$

In such a case, $\Gamma^+$ is always "not worse" than $\Gamma^-$ as it contains more expert data.

## 4 Learning frameworks for Solving VPIL Problems

In this section, we analyze the core challenge in VPIL problems with the pairwise datasets $\Gamma^+$ and $\Gamma^-$, i.e., recovering the occupancy measure of the expert policy $p_{\pi_\mathsf{E}}$. To this end, we propose two learning frameworks for VPIL with known $\alpha$ and unknown $\alpha$, respectively.

### 4.1 VPIL with Known $\alpha$

Most of the IL methods based on the occupancy measure need to assume that the demonstrations are sampled only from the expert policy, so that they can directly estimate $p_{\pi_\mathsf{E}}$ and match the learner's distribution with $p_{\pi_\mathsf{E}}$. However, the occupancy measure of the expert policy $p_{\pi_\mathsf{E}}$ is inaccessible in the VPIL problem, since both $\Gamma^+$ and $\Gamma^-$ contain non-expert data while the label of each data is unavailable. Below, we attempt to approximate $p_{\pi_\mathsf{E}}$ with $\{\Gamma^+, \Gamma^-\}$.

Let $n_+ = |\Gamma^+|$; $\widehat{p}_i^+(s,a) = (1-\gamma)\sum_{t=0}^{\infty} \gamma^t \mathbb{1}\left[(s_{t,i}, a_{t,i}) = (s,a)\right]$ for $i = 1, \ldots, n_+$ be the empirical occupancy measure of a trajectory $\tau_i \in \Gamma^+$, where $(s_{t,i}, a_{t,i}) \in \tau_i$ is a state action pair at time $t$. Let $\widehat{p}_{\pi_+}(s,a) = \sum_{i=1}^{n_+} \widehat{p}_i^+(s,a)/n_+$. Define $\widehat{p}_{\pi_-}(s,a)$ similarly. We have the following theorem.

**Theorem 2.** *Assume the pairwise datasets $(\Gamma^+, \Gamma^-)$ are generated following the procedure in Section 3.2. Let $p_{\pi_+}(s,a) = \mathbb{E}_{\Gamma^+}\left[\widehat{p}_{\pi_+}(s,a)\right]$ and $p_{\pi_-}(s,a) = \mathbb{E}_{\Gamma^-}\left[\widehat{p}_{\pi_-}(s,a)\right]$ be the expected occupancy measures of $\Gamma^+$ and $\Gamma^-$, where the randomness is taken over the draws of $\Gamma^+$ and $\Gamma^-$. Then, we have*

$$\begin{cases} p_{\pi_+}(s,a) = & \left(2\alpha - \alpha^2\right) p_{\pi_E}(s,a) + (1-\alpha)^2 p_{\pi_{NE}}(s,a), \\ p_{\pi_-}(s,a) = & \alpha^2 p_{\pi_E}(s,a) + (1-\alpha^2) p_{\pi_{NE}}(s,a), \end{cases} \tag{2}$$

$$\begin{cases} p_{\pi_E}(s,a) = & \frac{1+\alpha}{2\alpha}p_{\pi_+}(s,a) - \frac{1-\alpha}{2\alpha}p_{\pi_-}(s,a), \\ p_{\pi_{NE}}(s,a) = & -\frac{\alpha}{2(1-\alpha)}p_{\pi_+}(s,a) + \frac{2-\alpha}{2-2\alpha}p_{\pi_-}(s,a). \end{cases} \tag{3}$$

A proof can be found in Appendix A. The Theorem provides a feasible way to recover the unknown occupancy measure of the expert policy from contaminated data pools $\Gamma^+$ and $\Gamma^-$. Thus, we can use an off-the-shelf IL method to learn the policy.

Once we have recovered $p_{\pi_E}$ by Eq. (10), we can use an off-the-shelf IL method to learn the policy.

## 4.2 VPIL with Unknown $\alpha$

When facing a more challenging problem in which the expert ratio $\alpha$ is unknown, a straightforward way is to estimate the ratio first and then reconstruct the expert policy by the approach developed for known $\alpha$ cases. Here, we will first introduce how to estimate the expert ratio by reducing it into a mixture proportion estimation (MPE) problem [28] and then identify that a direct application of the estimated ratio is not accurate enough to reconstruct the expert policy. To this end, we further propose an iterative sample selection procedure to exploit the estimated expert ratio, which finally leads to a better approximation of the expert policy.

**Estimation of the Expert Ratio $\alpha$.** In this paragraph, we show that the estimation of the expert ratio $\alpha$ can be reduced to two MPE problems [28]. Let $\mathcal{P}$ and $\mathcal{N}$ be two probability distributions, and

$$\mathcal{U} = \beta\mathcal{P} + (1-\beta)\mathcal{N} \tag{4}$$

be a mixture of them with a certain proportion $\beta \in (0,1)$. The MPE problem studies how to estimate mixture proportion $\beta$ with the empirical observations sampled from $\mathcal{U}$ and $\mathcal{P}$ (not from $\mathcal{N}$). Over the decades, various algorithms were proposed with sound theoretical and empirical studies [31, 32].

To see how the estimation of the expert ratio $\alpha$ is related to the MPE problem, we rewrite (2) as

$$p_{\pi_+}(s,a) = \beta_1 p_{\pi_-}(s,a) + (1-\beta_1)p_{\pi_E} \quad \text{and} \quad p_{\pi_-}(s,a) = \beta_2 p_{\pi_+}(s,a) + (1-\beta_2)p_{\pi_{NE}}, \tag{5}$$

where $\beta_1 = \frac{1-\alpha}{1+\alpha}$ and $\beta_2 = \frac{\alpha}{2-\alpha}$. By taking $p_{\pi_+}$ as $\mathcal{U}$ and $p_{\pi_-}$ as $\mathcal{P}$, the first line of (5) shares the same formulation as the MPE problem (4). Since $p_{\pi_+}$ and $p_{\pi_-}$ are empirically accessible via $\Gamma^+$ and $\Gamma^+$, we can estimate $\beta_1$ by taking $\Gamma^+$ and $\Gamma^-$ as the input of any MPE solver and obtain the expert ratio by $\alpha = \frac{1-\beta_1}{1+\beta_1}$. The same argument also holds for the second line of (5), where we can take $p_{\pi_-}$ as $\mathcal{U}$ and $p_{\pi_+}$ as $\mathcal{P}$ to estimate $\beta$. Then, the expert ratio can also be obtained by $\alpha = \frac{2\beta_2}{1+\beta_2}$.

One might worry about the identifiability of the expert ratio $\alpha$ by estimating it with MPE techniques [28]. We can show that the true parameter is identifiable if the distribution $p_{\pi_+}$ and distribution $p_{\pi_-}$ are mutually irreducible [16] as follows:

**Proposition 1.** *Suppose the distributions $p_{\pi_E}$ and $p_{\pi_{NE}}$ are mutually irreducible such that there exists no decomposition of the form $p_{\pi_E} = (1-\eta)Q + \eta p_{\pi_{NE}}$ and $p_{\pi_{NE}} = (1-\eta')Q' + \eta' p_{\pi_E}$ for any probability distributions $Q, Q'$ and scalars $\eta, \eta' \in (0,1]$. Then, the true mixture proportions $\beta_1$ and $\beta_2$ are unique and can be identified by*

$$\begin{cases} \beta_1 = \sup\{\eta | p_{\pi_+} = \eta p_{\pi_-} + (1-\eta)K, K \in \mathcal{C}\}, \\ \beta_2 = \sup\{\eta' | p_{\pi_-} = \eta' p_{\pi_+} + (1-\eta')K', K' \in \mathcal{C}\}, \end{cases} \tag{6}$$

*where $\mathcal{C}$ is the set containing all possible probability distributions. Thus, $\alpha$ is identifiable by*

$$\alpha = (1-\beta_1)/(1+\beta_1) \quad or \quad \alpha = 2\beta_2/(1+\beta_2). \tag{7}$$

Proposition 1 demonstrates that the true mixture proportion $\beta_1$ (resp. $\beta_2$) can be identified by finding the maximum proportion of $p_{\pi_-}$ contained in $p_{\pi_+}$ (resp. $p_{\pi_+}$ contained in $p_{\pi_-}$). This idea can be empirically implemented via existing MPE methods [28, 33]. We note that the expert ratio is identifiable by either estimating $\beta_1$ or $\beta_2$ when we have infinite samples. In the finite sample case, the two estimators coming from different distribution components could lead to different estimation biases. Since the MPE solutions tend to have a positive estimation bias on the true value $\beta_1$ and $\beta_2$ as shown by [28, Theorem 12] and [31, Corollary 1], the estimation $\alpha = (1-\beta_1)/(1+\beta_1)$ tends to yield an underestimated $\alpha$ while that with $\beta_2$ will lead to an overestimation. Besides, when the number of expert data is quite small, the underlying true parameter $\beta_2$ would also have a small value,

---

**Algorithm 1** ExpertRatioEstimation

---

**Input:** Pairwise Demonstrations $\Gamma^+, \Gamma^-$; Numbers of Iterations $I$.
**Output:** Estimated Expert's Ratio $\widehat{\alpha}$.
 1: **function** ESTIMATION($\Gamma^+, \Gamma^-$)
 2:     Initialize $\mathcal{D}^+ \leftarrow \Gamma^+$ and $\mathcal{D}^- \leftarrow \Gamma^-$
 3:     **for** each iteration until $I$ **do**
 4:         Train a binary classifier $f_\psi : \mathcal{S} \times \mathcal{A} \rightarrow [0, 1]$ by taking $\mathcal{D}^+$ as positive data and $\mathcal{D}^-$ as negative.
 5:         Assign a score for each data $S^+ \leftarrow \{f_\psi(s, a) \mid (s, a) \in \mathcal{D}^+\}$ and $S^- \leftarrow \{f_\psi(s, a) \mid (s, a) \in \mathcal{D}^-\}$.
 6:         Estimate $\beta$ with a mixture proportion estimator by taking $S^+$ as $\mathcal{U}$ and $S^-$ as $\mathcal{P}$ in (4).
 7:         Estimate $\widehat{\alpha}$ by (7).                    ▷ We choose $(1 - \beta)/(1 + \beta)$ for better estimation.
 8:         Data selection for $\Gamma^+$: $\mathcal{D}^+ \leftarrow \{2\widehat{\alpha} - \widehat{\alpha}^2$ fraction of $\Gamma^+$ with top scores $S^+\}$.
 9:         Data selection for $\Gamma^-$: $\mathcal{D}^- \leftarrow \{1 - \widehat{\alpha}^2$ fraction of $\Gamma^-$ with bottom scores $S^-\}$.
10:     **end for**
11:     **return** $\widehat{\alpha}$
12: **end function**

---

---

**Algorithm 2** COMPILER/COMPILER-E

---

**Input:** Pairwise Demonstrations $\Gamma^+, \Gamma^-$; Expert ratio $\alpha$ (Unknown for COMPILER-E); Environment env.
**Output:** The learner policy $\pi_\theta$.
 1: Initialize the learner policy $\pi_\theta$ and the discriminator $D_\omega$.
 2: **if** $\alpha$ is unknown **then**                              ▷ COMPILER-E
 3:     Obtain the estimated expert ratio $\alpha \leftarrow$ ExpertRatioEstimation($\Gamma^+, \Gamma^-$).
 4: **else**                                                          ▷ COMPILER
 5:     $\alpha \leftarrow$ input($\alpha$).
 6: **end if**
 7: **for** each training steps **do**
 8:     Sample a batch of learner's data $(s, a) \sim p_{\pi_\theta}$ by the interactions between $\pi_\theta$ and env.
 9:     Sample a batch of demonstrations data $(s, a) \sim \widehat{p}_{\pi_+}$ and $(s, a) \sim \widehat{p}_{\pi_-}$ from $\Gamma^+$ and $\Gamma^-$.
10:     Update $D_\omega$ by maximizing (8).
11:     Update $\pi_\theta$ with $(s, a) \sim p_{\pi_\theta}$ and the reward $-\log D_\omega(s, a)$ using off-the-shelf RL algorithm.
12: **end for**

---

and its estimation would be highly unstable. Thus, we choose to estimate $\alpha$ with $\beta_1$ in our algorithm. The empirical studies in Section 6.3 also supported our choice.

The relationship (2) between the expert and non-expert data shares a similar formulation to that of the unlabeled-unlabeled (UU) classification [34], which is a specific kind of weakly supervised learning problem studying how to train a classifier from two unlabeled datasets and requires the knowledge of mixture proportions. However, how to estimate these mixture proportions is still an open problem in the weakly supervised learning literature. Although our reduction is developed for estimating the expert ratio for IL problems, it can be applied to a UU learning setting for independent interest.

**Reconstruct Expert Policy by Data Selection.** To handle the MPE task for high dimensional data, a practice is to first train a classifier with probability output and then conduct the MPE on the probability outputs of samples [35]. However, the quality of the trained classifier turns out to depend on the estimation accuracy of $\alpha$. On the other hand, we found that it is possible to filter out the undesired component of the pairwise datasets ($\widehat{p}_{\pi_E}$ in $\Gamma^-$ and $\widehat{p}_{\pi_{NE}}$ in $\Gamma^+$) and keep the desired data to enhance estimating $\alpha$. Therefore, also inspired by [31, 36] of the distribution shift problem and weakly supervised learning, we propose an iteration-based learning framework. In each iteration, we throw away the non-expert data with higher confidence in $\Gamma^+$ and the expert data with higher confidence in $\Gamma^-$ after estimating $\alpha$, and train a classifier with the datasets after selection. The detailed learning process can be found in Algorithm 1.

After we obtained the estimated $\widehat{\alpha}$, we can recover $p_{\pi_E}$ as by Eq. (10).

# 5 COMParative Imitation LEarning with Risk rewriting (COMPILER)

We have illustrated how to estimate the occupancy measure of the expert policy $p_{\pi_E}$ from $(\Gamma^+, \Gamma^-)$ under known and unknown $\alpha$ respectively. Here we describe how to adopt these learning frameworks into end-to-end algorithms for learning an policy. We name our algorithm for VPIL with known $\alpha$ **COMParative Imitation LEarning with Risk-rewriting (COMPILER)**, and with unknown $\alpha$ **COMParative Imitation LEarning with Risk-rewriting by Estimation (COMPILER-E)**.

Current state-of-the-art adversarial IL methods [37, 1] aim to learn a policy by matching the occupancy measure between the learner and the expert policies. We fuse our methods with one of the representative adversarial IL methods, Generative Adversarial Imitation Learning (GAIL) [1], whose optimization problem is given as follows:

$$\min_{\theta \in \Theta} \max_{w \in \mathcal{W}} \mathbb{E}_{(s,a) \sim p_{\pi_\theta}} \log D_w(s, a) + \mathbb{E}_{(s,a) \sim p_{\pi_E}} \log(1 - D_w(s, a)),$$

where $D_w : \mathcal{S} \times \mathcal{A} \mapsto [0, 1]$ parameterized by $w$ is a discriminator. $p_{\pi_\theta}$ parameterized by $\theta$ is the occupancy measure of a policy. In our setting, the expert demonstration of $p_{\pi_E}$ is unavailable.

**COMPILER.** As suggested by Theorem 2, we can recover $p_{\pi_E}$ with the occupancy measures of $\Gamma^+$ and $\Gamma^-$. Then, we can train the policy $p_{\pi_\theta}$ by mincing $p_{\pi_E} = \frac{1+\alpha}{2\alpha} p_{\pi_+} - \frac{1-\alpha}{2\alpha} p_{\pi_-}$ in Eq. (10). Specifically, plugging (2) into (8) and approximating $p_{\pi_+}$ and $p_{\pi_-}$ with their empirical versions, we can obtain the desirable target. However, such rewriting can introduce a negative term and lead to overfitting [38, 39]. To avoid such an undesired phenomenon, instead of training the policy $\pi_\theta$ to make $p_{\pi_\theta}$ match $\frac{1+\alpha}{2\alpha} \widehat{p}_{\pi_+} - \frac{1-\alpha}{2\alpha} \widehat{p}_{\pi_-}$, we twist the objective function of GAIL (8) to minimize the discrepancy between $\frac{2\alpha}{1+\alpha} p_{\pi_\theta} + \frac{1-\alpha}{1+\alpha} \widehat{p}_{\pi_-}$ and $\widehat{p}_{\pi_+}$, which gives the following optimization problem without the negative term:

$$\min_{\theta \in \Theta} \max_{w \in \mathcal{W}} 2\alpha \mathbb{E}_{(s,a) \sim p_{\pi_\theta}} \log D_w(s, a) + (1 - \alpha) \mathbb{E}_{(s,a) \sim \widehat{p}_{\pi_-}} \log(D_w(s, a))$$
$$+ (1 + \alpha) \mathbb{E}_{(s,a) \sim \widehat{p}_{\pi_+}} \log(1 - D_w(s, a)). \tag{8}$$

Let $\widehat{V}(\pi_\theta, D_w)$ be the objective function in (8) and $V(\pi_\theta, D_w)$ be its expectation established on $p_{\pi_+}$ and $p_{\pi_-}$. We have the following theorem for the estimation error of the discriminator trained by (8).

**Theorem 3.** *Let $\mathcal{W}$ be a parameter space for training the discriminator and $D_{\mathcal{W}} = \{D_w \mid w \in \mathcal{W}\}$ be the hypothesis space. Assume the functions $|\log D_w(s, a)| \leq B$ and $|\log(1 - D_w(s, a))| \leq B$ are upper-bounded for any state-action pair $(s, a) \in \mathcal{S} \times \mathcal{A}$ and $w \in \mathcal{W}$. Further assume both the functions $\log D_w(s, a)$ and $\log(1 - D_w(s, a))$ are L-Lipschitz continuous in the state-action space. For a fixed policy $\pi_\theta$, let $\Gamma^\theta = \{\tau_i^\theta\}_{i=1}^{n_\theta}$ be trajectories generated from $\pi_\theta$. Then, for any $\delta \in (0, 1)$, with probability at least $1 - \delta$, we have*

$$V(\pi_\theta, D_w^*) - V(\pi_\theta, \widehat{D}_w) \leq 4L(1 + \alpha)\mathcal{R}_{n_+}(D_{\mathcal{W}}) + 4L(1 - \alpha)\mathcal{R}_{n_-}(D_{\mathcal{W}}))$$
$$+ 8L\mathcal{R}_{n_\theta}(D_{\mathcal{W}}) + C(\delta) \left( \frac{1}{\sqrt{n_\theta}} + \frac{1}{\sqrt{n_+}} + \frac{1}{\sqrt{n_-}} \right),$$

*where $\widehat{D}_w = \arg\max_{w \in \mathcal{W}} \widehat{V}(\pi_\theta, D_w)$ and $D_w^* = \arg\max_{w \in \mathcal{W}} V(\pi_\theta, D_w)$. The constants $n_+$ and $n_-$ are the number of trajectories in $\Gamma^+$ and $\Gamma^-$. We define $C(\delta) = 4B\sqrt{\log(6/\delta)}$. The empirical Radamacher complexities [40] on datasets $\Gamma^+$, $\Gamma^-$, and $\Gamma^\theta$ are denoted by $\mathcal{R}_{n_+}$, $\mathcal{R}_{n_-}$, and $\mathcal{R}_{n_\theta}$.*

A proof is given in Appendix A. The theorem shows that the discriminator trained by (8) converges to the one optimized with the true distribution $p_{\pi_+}$ and $p_{\pi_-}$ at each step of the training.

**COMPILER-E.** For the VPIL problem with unknown $\alpha$, first we estimate $\widehat{\alpha}$ by Algorithm 1, then we learn the policy by (8) with $\widehat{\alpha}$.

We integrate the two algorithmic processes COMPILER and COMPILER-E into Algorithm 2.

# 6 Experiments

In this section, we conducted extensive experiments under the setting of VPIL. Through the experiments, we want to investigate the following questions: (1) Can COMPILER and COMPILER-E solve VPIL problems under various expert ratios in demonstrations? (2) Is COMPILER-E still valid when using different MPE estimators to obtain $\widehat{\alpha}$? (3) How is the $\alpha$ estimation with $\beta_1$ and $\beta_2$ as in (7)?

Table 1: The detailed information of demonstrations in the empirical studies.

|  | Dimension of $\mathcal{S}$ | Dimension of $\mathcal{A}$ | Non-Expert 1 | Non-Expert 2 | Expert | $\alpha$ |
|---|---|---|---|---|---|---|
| Hopper | 11 | 3 | 1142.16±159.28 | 1817.80±819.69 | 3606.11±43.95 | {0.1, 0.2, 0.3, 0.4, 0.5} |
| Swimmer | 8 | 2 | 46.28±1.87 | 62.64±16.82 | 100.63±4.69 | {0.1, 0.2, 0.3, 0.4, 0.5} |
| Walker2d | 17 | 6 | 410.98±56.91 | 766.67±398.85 | 3204.37±848.55 | {0.1, 0.2, 0.3, 0.4, 0.5} |
| HalfCheetah | 17 | 6 | 532.50±58.66 | 864.18±335.90 | 1599.06±41.97 | {0.1, 0.2, 0.3, 0.4, 0.5} |

## 6.1 Setup

To investigate the answer to the first question, we chose 20 different VPIL tasks with four MuJoCo benchmark environments to evaluate the performance of the contenders and our approaches under five different $\alpha$ levels. The detailed setups are reported as follows.

**Environments and Demonstrations.** We set *HalfCheetah, Hopper, Swimmer*, and *Walker2d* in MuJoCo as the basic environments. For each experiment, the demonstration pool contains 100 trajectories with different expert ratios $\alpha = \{0.1, 0.2, 0.3, 0.4, 0.5\}$, in which $\alpha = 0.1$ is the most difficult situation of the VPIL problem. We also trained an expert-level RL agent and two non-expert RL agents as demonstrators by DDPG algorithm [41]. The details of the environment and the demonstrations can be found in Table 1.

**Dataset generation process.** We start with the demonstration pool $\{\tau_i\}_{i=1}^{N}$, which contains $\alpha N$ trajectories sampled from the optimal policy $\pi_{\mathsf{E}}$ and $(1 - \alpha)N$ trajectories from the non-optimal policy $\pi_{\mathsf{NE}}$. For notation simplicity, we denote by $\mathcal{D}_+$ the part containing expert demonstrations only and by $\mathcal{D}_-$ the part for the non-expert demonstrations. Then, we can generate the dataset as follows,

$$\begin{cases} \Gamma^+ = & \mathrm{sample}(\mathcal{D}_+, (2\alpha - \alpha^2)N) \bigcup \mathrm{sample}(\mathcal{D}_-, (1 - \alpha)^2 N), \\ \Gamma^- = & \mathrm{sample}(\mathcal{D}_+, \alpha^2 N) \bigcup \mathrm{sample}(\mathcal{D}_-, (1 - \alpha^2)N), \end{cases}$$

where $\mathrm{sample}(\mathcal{D}, N)$ is the function that samples $N$ trajectories from the dataset $\mathcal{D}$.

**Contenders.** Since $\Gamma^+$ contains more expert demonstrations, here we use Behavior Cloning [42], GAIL [1], and AIRL [43] with $\Gamma^+$ only as basic baselines. Also, we provide GAIL with expert-level demonstrations (GAIL_expert) as the skyline of all methods. Besides, we set T-REX [13], a state-of-the-art preference-based IL method, by taking that every data in $\Gamma^+$ is more expert-like than that in $\Gamma^-$ as a form of preference to train its reward function. We also set the variants of T-REX algorithms, D-REX [22] and SSRR [44], that directly generate the ranking information through the imperfect datasets. CAIL proposed a confidence-based method to imitate from mixed demonstrations [14]. In the experiment, we provide 5% trajectory labels to meet its requirement as suggested in their paper. Each method ran 4e7 steps. 5 trials were conducted with different seeds for each task.

Meanwhile, to investigate the answer to the second question, we conducted Best Bin Estimation (BBE) [31] and Kernel Mean (KM) [28] algorithm to estimate $\beta$ in algorithm 1 of COMPILER-E, as COMPILER-E (BBE) and COMPILER-E (KM) respectively. We note that COMPILER-E cannot obtain the true $\alpha$ during the whole experiment.

**Implementation of $\alpha$ estimation.** We implement a four-layer fully connected neural network as the binary classifier $f_\psi$ in Algorithm 1. The neurons in each layer are 1000, 1000, 100, and 50 respectively. The activation function of each layer is ReLU, and the output value will go through the sigmoid function to obtain the score for $\Gamma^+$ and $\Gamma^-$ as $S^+$ and $S^-$ respectively. The optimizer for $f_\psi$ is stochastic gradient descent. We totally train $f_\psi$ for 1000 epochs with bath size 1024. The initial learning rate is 5e-4, with an exponential decay rate of 0.99 at each step.

**Implementation of policy training.** We choose Proximal Policy Optimization (PPO) [45] as the basic RL algorithm, and set all hyper-parameters, update frequency, and network architectures of the policy part the same as [46]. Besides, the hyper-parameters of the discriminator for all methods were the same: The discriminator was updated using Adam with a decayed learning rate of $3 \times 10^{-4}$; the batch size was 256. The ratio of update frequency between the learner and discriminator was 3: 1. For T-REX algorithm [13], we use the codes with default hyperparameters and model architecture of their official implementation.

## 6.2 Empirical Results

The final results are gathered in Table 2. We can see that since AIRL and GAIL are standard IL algorithms, they cannot adaptively filter out non-expert data, so the underlying non-expert data reduces the performance of them. T-REX did not obtain promising results on VPIL tasks, which

Table 2: The episodic returns of each method on different tasks with five random seeds.

| Algorithm | HalfCheetah | | | | | Hopper | | | | |
| --- | --- | --- | --- | --- | --- | --- | --- | --- | --- | --- |
| | $\alpha = 0.1$ | $\alpha = 0.2$ | $\alpha = 0.3$ | $\alpha = 0.4$ | $\alpha = 0.5$ | $\alpha = 0.1$ | $\alpha = 0.2$ | $\alpha = 0.3$ | $\alpha = 0.4$ | $\alpha = 0.5$ |
| GAIL_expert | $1194.69 \pm 9.39$ | $1183.79 \pm 10.94$ | $1149.63 \pm 10.78$ | $1175.55 \pm 5.03$ | $1178.95 \pm 9.89$ | $3205.43 \pm 53.78$ | $3357.21 \pm 118.81$ | $3009.00 \pm 126.88$ | $2520.26 \pm 79.07$ | $3207.89 \pm 272.30$ |
| Behavior Cloning | $-252.99 \pm 40.41$ | $-280.13 \pm 12.95$ | $-259.96 \pm 15.01$ | $-233.14 \pm 27.89$ | $-248.84 \pm 57.95$ | $3.37 \pm 1.00$ | $5.03 \pm 0.39$ | $4.34 \pm 1.04$ | $3.56 \pm 0.70$ | $6.14 \pm 2.47$ |
| GAIL | $862.44 \pm 11.47$ | $638.34 \pm 211.12$ | $663.98 \pm 227.04$ | $798.94 \pm 201.16$ | $833.82 \pm 199.45$ | $2889.95 \pm 108.94$ | $2248.22 \pm 738.06$ | $2687.68 \pm 395.29$ | $3068.24 \pm 184.86$ | $3058.85 \pm 408.20$ |
| AIRL | $767.28 \pm 14.40$ | $749.36 \pm 4.36$ | $750.17 \pm 13.53$ | $824.88 \pm 13.44$ | $948.51 \pm 20.62$ | $1689.21 \pm 158.41$ | $2350.81 \pm 602.29$ | $2430.85 \pm 176.82$ | $3120.07 \pm 146.55$ | $2930.19 \pm 214.47$ |
| T-REX | $-307.86 \pm 84.96$ | $-424.82 \pm 232.80$ | $-595.05 \pm 30.70$ | $-507.48 \pm 10.59$ | $-559.07 \pm 47.79$ | $2777.79 \pm 258.43$ | $2645.89 \pm 148.47$ | $2567.30 \pm 264.91$ | $2764.23 \pm 23.34$ | $2526.49 \pm 390.17$ |
| SSRR | $-23.12 \pm 432.45$ | $636.43 \pm 132.53$ | $644.47 \pm 221.43$ | $532.41 \pm 146.23$ | $812.43 \pm 346.32$ | $812.25 \pm 547.45$ | $743.92 \pm 12.43$ | $834.14 \pm 54.82$ | $1632.65 \pm 72.85$ | $2093.16 \pm 63.75$ |
| D-REX | $-36.04 \pm 231.24$ | $598.26 \pm 118.27$ | $498.92 \pm 158.82$ | $494.79 \pm 132.04$ | $750.46 \pm 302.44$ | $780.45 \pm 1064.72$ | $768.92 \pm 35.32$ | $783.24 \pm 40.63$ | $1489.07 \pm 50.02$ | $1775.78 \pm 79.66$ |
| CAIL | $729.30 \pm 14.38$ | $730.67 \pm 46.32$ | $727.16 \pm 39.90$ | $877.17 \pm 21.70$ | $1023.79 \pm 18.77$ | $1662.75 \pm 15.43$ | $2567.27 \pm 30.04$ | $2600.46 \pm 18.12$ | $3181.73 \pm 30.26$ | $\mathbf{3613.39 \pm 4.96}$ |
| COMPILER | $\mathbf{922.63 \pm 35.01}$ | $1022.26 \pm 8.09$ | $1108.86 \pm 5.77$ | $1158.81 \pm 21.99$ | $1176.47 \pm 4.89$ | $2968.06 \pm 142.26$ | $3140.12 \pm 182.90$ | $3342.82 \pm 170.21$ | $3318.97 \pm 132.90$ | $3369.32 \pm 93.55$ |
| COMPILER-E (BBE) | $893.46 \pm 13.85$ | $993.02 \pm 10.54$ | $1085.83 \pm 19.45$ | $1140.59 \pm 18.41$ | $1171.26 \pm 16.20$ | $\mathbf{3304.72 \pm 146.63}$ | $3321.49 \pm 71.11$ | $3432.43 \pm 89.40$ | $\mathbf{3398.16 \pm 164.11}$ | $3339.86 \pm 167.70$ |
| COMPILER-E (KM) | $901.67 \pm 17.15$ | $826.80 \pm 123.83$ | $911.47 \pm 119.29$ | $1062.98 \pm 9.37$ | $1075.86 \pm 81.26$ | $2828.93 \pm 215.48$ | $\mathbf{3459.24 \pm 49.37}$ | $\mathbf{3439.01 \pm 46.82}$ | $3175.60 \pm 80.51$ | $3402.89 \pm 71.79$ |

| Algorithm | Swimmer | | | | | Walker2d | | | | |
| --- | --- | --- | --- | --- | --- | --- | --- | --- | --- | --- |
| | $\alpha = 0.1$ | $\alpha = 0.2$ | $\alpha = 0.3$ | $\alpha = 0.4$ | $\alpha = 0.5$ | $\alpha = 0.1$ | $\alpha = 0.2$ | $\alpha = 0.3$ | $\alpha = 0.4$ | $\alpha = 0.5$ |
| GAIL_expert | $99.55 \pm 1.75$ | $102.65 \pm 1.78$ | $100.70 \pm 0.56$ | $100.52 \pm 1.70$ | $98.82 \pm 2.04$ | $3469.81 \pm 218.18$ | $3495.31 \pm 206.58$ | $3424.13 \pm 102.34$ | $3426.62 \pm 198.15$ | $3328.20 \pm 357.37$ |
| Behavior Cloning | $17.13 \pm 20.16$ | $51.33 \pm 11.61$ | $19.21 \pm 42.68$ | $39.40 \pm 37.49$ | $-10.26 \pm 23.34$ | $-5.82 \pm 7.54$ | $-12.33 \pm 2.05$ | $19.50 \pm 39.39$ | $-14.07 \pm 5.33$ | $-6.27 \pm 13.46$ |
| GAIL | $75.11 \pm 4.25$ | $60.96 \pm 11.66$ | $88.21 \pm 12.46$ | $90.34 \pm 8.28$ | $96.21 \pm 4.99$ | $2883.94 \pm 230.93$ | $2779.58 \pm 792.39$ | $2751.23 \pm 711.34$ | $2914.64 \pm 530.28$ | $2936.05 \pm 604.15$ |
| AIRL | $31.66 \pm 3.52$ | $50.44 \pm 5.73$ | $73.72 \pm 4.33$ | $83.65 \pm 6.77$ | $94.55 \pm 4.62$ | $2003.95 \pm 272.55$ | $2849.55 \pm 85.02$ | $2914.45 \pm 78.68$ | $2970.29 \pm 149.52$ | $3009.35 \pm 236.45$ |
| T-REX | $7.07 \pm 37.40$ | $0.57 \pm 0.66$ | $14.80 \pm 4.46$ | $6.29 \pm 10.48$ | $0.70 \pm 24.89$ | $1007.39 \pm 217.20$ | $1660.39 \pm 541.72$ | $1845.43 \pm 695.87$ | $1930.96 \pm 660.01$ | $2091.45 \pm 314.22$ |
| SSRR | $50.44 \pm 3.53$ | $53.42 \pm 6.64$ | $76.41 \pm 1.64$ | $80.14 \pm 6.52$ | $83.15 \pm 12.98$ | $629.42 \pm 8.63$ | $943.26 \pm 73.36$ | $985.25 \pm 15.68$ | $1534.02 \pm 86.42$ | $1620.73 \pm 119.80$ |
| D-REX | $55.04 \pm 3.70$ | $41.32 \pm 10.65$ | $73.22 \pm 11.72$ | $59.92 \pm 5.57$ | $63.22 \pm 11.72$ | $558.65 \pm 10.59$ | $803.20 \pm 15.15$ | $873.89 \pm 32.41$ | $961.01 \pm 25.84$ | $779.17 \pm 19.72$ |
| CAIL | $49.53 \pm 1.33$ | $66.94 \pm 12.15$ | $78.15 \pm 6.01$ | $95.88 \pm 1.87$ | $\mathbf{102.12 \pm 1.14}$ | $2023.80 \pm 123.59$ | $3022.30 \pm 25.46$ | $3260.85 \pm 103.69$ | $3530.55 \pm 165.45$ | $3164.23 \pm 36.05$ |
| COMPILER | $\mathbf{99.03 \pm 0.98}$ | $101.34 \pm 1.72$ | $100.87 \pm 1.21$ | $\mathbf{100.86 \pm 2.05}$ | $100.01 \pm 0.26$ | $3342.64 \pm 172.10$ | $3516.76 \pm 132.31$ | $3400.51 \pm 112.84$ | $3536.91 \pm 166.69$ | $\mathbf{3668.58 \pm 115.10}$ |
| COMPILER-E (BBE) | $89.86 \pm 5.23$ | $\mathbf{102.43 \pm 0.70}$ | $\mathbf{101.26 \pm 2.09}$ | $98.78 \pm 3.51$ | $101.69 \pm 0.41$ | $3365.31 \pm 106.52$ | $\mathbf{3621.73 \pm 69.74}$ | $\mathbf{3583.22 \pm 15.59}$ | $\mathbf{3543.71 \pm 66.53}$ | $3649.94 \pm 57.41$ |
| COMPILER-E (KM) | $93.30 \pm 2.10$ | $91.79 \pm 10.91$ | $100.17 \pm 3.15$ | $98.82 \pm 1.52$ | $101.88 \pm 0.69$ | $\mathbf{3467.94 \pm 44.26}$ | $3444.01 \pm 223.64$ | $3514.53 \pm 183.36$ | $3491.87 \pm 343.31$ | $3629.15 \pm 73.66$ |

verifies that vague pairwise comparison can not be considered as full-ranking information, so the preference-based method cannot be used to tackle such problems. In addition, both SSRR and D-REX are updated versions of T-REX using noise to generate ranking information. They still cannot achieve good results due to its inability to address the core challenges of the VPIL problem. This phenomenon further demonstrates that preference-based algorithms cannot solve VPIL problems without explicit preference information, and some assumptions may not be held in VPIL problems. As $\alpha$ grows, we can see that the performances of almost all the contenders increased more or less, except for the skyline GAIL_expert, while COMPILER and COMPILER-E got the biggest performance boost and achieved the best performances. Even under the $\alpha = 0.1$ situation, COMPILER and COMPILER-E have achieved expert-level performance on Hopper and Walker2d environments. For CAIL, it shows competitive performance with COMPILER and COMPILER-E under high $\alpha$ situations, but still falls short in low $\alpha$ cases. The reason for this phenomenon is that when $\alpha$ is relatively low, the trajectory quality corresponding to the provided trajectory labels may also be poor. Under such circumstances, estimating confidence is extremely difficult. As a result, CAIL generally performs suboptimally in low $\alpha$ cases. On the other hand, our algorithms can achieve comparable results with CAIL in high alpha scenarios. This indicates that our methods reasonably utilize the structure of the VPIL problem and can achieve results on par with the state-of-the-art algorithm without introducing additional information. Meanwhile, COMPILER (BBE) and COMPILER (KM) both achieved promising and comparable performances, which indicates that COMPILER-E is a general algorithm and can be operated with any effective MPE method.

### 6.3 Case Study: Estimation Effect on Data Selection

**Overestimation and underestimation comparisons.** To investigate if the superiority of COMPILER-E comes from precisely estimating $\alpha$, and also to answer the third question about using two different estimations in (7) in Section 6, we reported the estimation curves under different true $\alpha$ on HalfCheetah with BBE and KM estimators, as shown in Figure 3. We can see that the overestimation method obtained $\widehat{\alpha}$ that far exceeded the true $\alpha$, especially when $\alpha$ was smaller; while the underestimation method was very accurate despite obtaining a slightly lower $\widehat{\alpha}$ value compared to the true $\alpha$, for both BBE and KM estimators. The results also suggested that we indeed need to use the underestimation method for COMPILER-E when the expert ratio $\alpha$ is small. It also reflects that the power of COMPILER-E for solving VPIL problems indeed comes from accurately estimating the $\alpha$ value. More environmental results can be found in Appendix B.

**Thrown data effect.** To further investigate what caused the performance gap between the overestimation and underestimation, we analyze the effect on the proportion of data selections in Algorithm 1. Also connected to the expert and non-expert proportions of $\Gamma^+$ and $\Gamma^-$ in (2), we can calculate the theoretical value of the ratio differences of thrown data under different estimations with that under true $\alpha$, as shown in Figure 3.

We can see that the overestimated method threw away more data in $\Gamma^-$ and fewer data in $\Gamma^+$ (corresponding to the upper left triangles of the red boxes in the figures); meanwhile, the underestimated method did the opposite (corresponding to the lower right triangles). However, as analyzed in Section 4.2, the overestimated method relied heavily on $p_{\pi_E}$ of $\Gamma^-$ dataset, whose components, were quite small. In the case of finite data, once $p_{\pi_E}$ of $\Gamma^-$ were overthrown, the remaining part of $p_{\pi_E}$ became less, making the estimation further high, which led to a vicious circle. The phenomenon was especially severe in the case of low $\alpha$. This is the reason why $\widehat{\alpha}$ of the overestimated method became

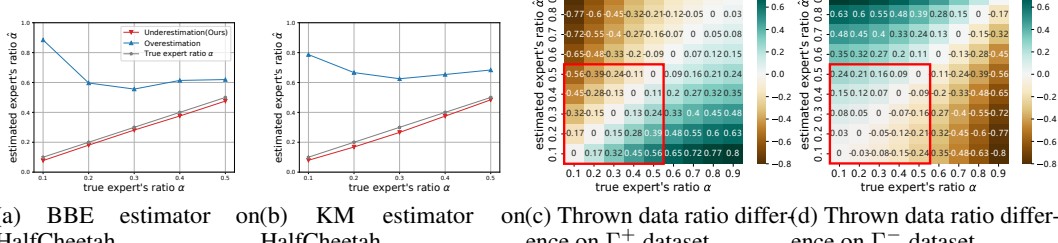

(a) BBE estimator on HalfCheetah  (b) KM estimator on HalfCheetah  (c) Thrown data ratio difference on $\Gamma^+$ dataset  (d) Thrown data ratio difference on $\Gamma^-$ dataset

Figure 3: (a, b) The comparisons of overestimated ($\frac{2\beta}{1+\beta}$) and underestimated ($\frac{1-\beta}{1+\beta}$) methods on various tasks with BBE and KM estimators. (c,d) The distribution map of thrown data ratio using estimated $\widehat{\alpha}$ and true $\alpha$. The thrown data ratio difference on $\Gamma^+$ is $(1-\widehat{\alpha})^2 - (1-\alpha)^2$, while that on $\Gamma^-$ is $\widehat{\alpha}^2 - \alpha^2$. The range marked by the red box is considered in our experiments, in which the expert ratio is smaller than (or equal to) the non-expert one.

bigger as true $\alpha$ decreased. On the other hand, although the underestimated method also overthrew the non-expert data in $\Gamma^+$, the proportion of these data was relatively high. So even if more were thrown away at the beginning, it will not affect the accuracy of the estimation. This is the reason why using $\frac{1-\beta}{1+\beta}$ to estimate $\widehat{\alpha}$ is relatively stable and accurate.

### 6.4 Case Study: Assessing the Robustness of the COMPILER Algorithm

To evaluate the robustness of the COMPILER algorithm, a comprehensive set of experiments were conducted. These were carried out in four distinct environments, implementing tasks with $\alpha = 0.5$. Each experiment was repeated five times to ensure the reliability of the results. To emulate real-world conditions, different levels of noise were introduced to the parameter $\alpha$, resulting in the generation of corresponding noisy datasets. This noise was denoted as $\varepsilon$, thereby resulting in $\widehat{\alpha} = \alpha + \varepsilon$.

Figure 4 presents the findings from these experiments. Remarkably, the COMPILER algorithm demonstrated consistent performance, achieving at least 96% of the standard performance even in the presence of various noise levels. This robustness, as exhibited in the results, underlines the capacity of COMPILER to effectively handle and perform under noisy conditions, specifically when the provided $\alpha$ is subject to disturbances. This emphasizes not only the resilience of the algorithm but also its potential for deployment in real-world situations where data is seldom perfect.

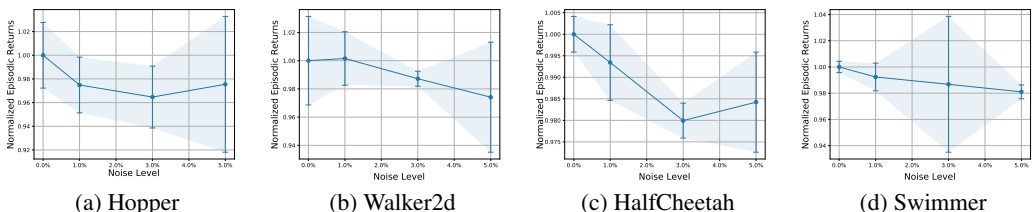

(a) Hopper  (b) Walker2d  (c) HalfCheetah  (d) Swimmer

Figure 4: The performance of COMPILER under different noise levels.

## 7 Conclusion

In this work, we formulated the problem of *Vaguely Pairwise Imitation Learning (VPIL)*, in which mixed expert and non-expert demonstrations are present, and the data collector only provides vague pairwise information of demonstrations. To solve this problem, we proposed two learning paradigms, with risk rewriting and mixture proportion estimations (MPE), to recover the expert distribution with the known expert ratio $\alpha$ and unknown one respectively. Afterward, we showed that these paradigms can be integrated with off-the-shelf IL methods, such as GAIL, to form the algorithm COMParative Imitation LEarning with Risk rewriting (COMPILER) and that by Estimation (COMPILER-E) to solve the VPIL problem with known and unknown $\alpha$ respectively. The experimental results showed that our methods outperformed standard and preference-based IL methods on a variety of tasks. In the future, we hope to use our algorithms to address more challenging problems, such as VPIL with multiple and noisy annotators.

## Acknowledgments

This research was supported by the Institute for AI and Beyond, UTokyo; JST SPRING, Grant Number JPMJSP2108. The authors would like to thank Johannes Ackermann and the anonymous reviewers for their insightful comments and suggestions.

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

# A  Proof for the Theorems

In this section, we provide the proof for Theorem 2, Proposition 1, and Theorem 3 in the main body of the paper.

**Theorem 2.** *Assume the pairwise datasets $(\Gamma^+, \Gamma^-)$ are generated following the procedure in Section 3.2 in the main paper. Let $p_{\pi_+}(s,a) = \mathbb{E}_{\Gamma^+}\left[\widehat{p}_{\pi_+}(s,a)\right]$ and $p_{\pi_-}(s,a) = \mathbb{E}_{\Gamma^-}\left[\widehat{p}_{\pi_-}(s,a)\right]$ be the expected occupancy measures of $\Gamma^+$ and $\Gamma^-$, where the randomness is taken over the draws of $\Gamma^+$ and $\Gamma^-$. Then, we have*

$$\begin{cases} p_{\pi_+}(s,a) = & \left(2\alpha - \alpha^2\right) p_{\pi_E}(s,a) + (1-\alpha)^2 p_{\pi_{NE}}(s,a), \\ p_{\pi_-}(s,a) = & \alpha^2 p_{\pi_E}(s,a) + (1-\alpha^2)p_{\pi_{NE}}(s,a), \end{cases} \tag{9}$$

$$\begin{cases} p_{\pi_E}(s,a) = & \frac{1+\alpha}{2\alpha}p_{\pi_+}(s,a) - \frac{1-\alpha}{2\alpha}p_{\pi_-}(s,a), \\ p_{\pi_{NE}}(s,a) = & -\frac{\alpha}{2(1-\alpha)}p_{\pi_+}(s,a) + \frac{2-\alpha}{2-2\alpha}p_{\pi_-}(s,a). \end{cases} \tag{10}$$

*Proof.* We first prove the first line of (9). Considering the data collection process introduced in Section 3.2, where a pair of trajectories $(\tau^{(1)}, \tau^{(2)})$ is independently sampled from the demonstrator pool. When one trajectory enjoys a better quality than another, the collector would put the better one into $\Gamma^+$. Otherwise, the two trajectories will be randomly allocated. Let $\widehat{p}_i^+(s,a) = (1-\gamma)\sum_{t=0}^{\infty} \gamma^t \mathbb{1}\left[(s_{t,i}, a_{t,i}) = (s,a)\right]$ be the empirical occupancy measure for the trajectory $\tau_i \in \Gamma^+$. Then, further define the even $E_1 = \{\tau_i \text{ is sampled from } \pi_E\}$ and event $E_1 = \{\tau_i \text{ is sampled from } \pi_{NE}\}$. We have

$$\mathbb{E}[\widehat{p}_i^+(s,a)] = \Pr[E_1]\mathbb{E}_{\Gamma^+}\left[\widehat{p}_i^+(s,a) \mid E_1\right] + \Pr[E_2]\mathbb{E}_{\Gamma^+}\left[\widehat{p}_i^+(s,a) \mid E_2\right]$$
$$= \Pr[E_1]p_{\pi_E} + \Pr[E_2]p_{\pi_{NE}}. \tag{11}$$

The second equality is due to $\tau_i$ is i.i.d. sampled from $p_{\pi_E}$ under event $E_1$ and is i.i.d. sampled from $p_{\pi_-}$ under event $E_2$. Then, let the even $A_1 = \{\text{both } \tau^{(1)}, \tau^{(2)} \text{ are sampled from } \pi_E\}$, $A_2 = \{\text{both } \tau^{(1)}, \tau^{(2)} \text{ are sampled from } \pi_{NE}\}$ and $A_3 = \{\text{other situation}\}$. Then, we have

$$\Pr[E_1] = \Pr[A_1]\Pr[E_1 \mid A_1] + \Pr[A_2]\Pr[E_1 \mid A_2] + \Pr[A_3]\Pr[E_1 \mid A_3]$$
$$= \alpha^2 + 2\alpha(1-\alpha) = 2\alpha - \alpha^2,$$

where the second inequality is due to $\Pr[E_1 \mid A_1] = \Pr[E_3 \mid A_3] = 1$ and $\Pr[E_1 \mid A_2] = 0$ according to the data collection process. We can also show the probability of event $E_2$ is $\Pr[E_1] = (1-\alpha)^2$, then we have

$$\mathbb{E}[\widehat{p}_i^+(s,a)] = \left(2\alpha - \alpha^2\right) p_{\pi_E}(s,a) + (1-\alpha)^2 p_{\pi_{NE}}(s,a).$$

Since $\widehat{p}_{\pi_+}(s,a) = \sum_{i=1}^{n_+} \widehat{p}_i^+(s,a)/n_+$, we have $\mathbb{E}[\widehat{p}_{\pi_+}(s,a)] = \sum_{i=1}^{n_+} \mathbb{E}[\widehat{p}_i^+(s,a)]/n_+ = \left(2\alpha - \alpha^2\right) p_{\pi_E}(s,a) + (1-\alpha)^2 p_{\pi_{NE}}(s,a)$, which completes the proof for the first equality of (9). The second equality of (9) can be proved following a similar arguments $\qquad\square$

**Proposition 2.** *Suppose the distributions $p_{\pi_E}$ and $p_{\pi_{NE}}$ are mutually irreducible such that there exists no decomposition of the form $p_{\pi_E} = (1-\eta)Q + \eta p_{\pi_{NE}}$ and $p_{\pi_{NE}} = (1-\eta')Q' + \eta' p_{\pi_E}$ for any probability distributions $Q, Q'$ and scalars $\eta, \eta' \in (0,1]$. Then, the true mixture proportions $\beta_1$ and $\beta_2$ are unique and can be identified by*

$$\begin{cases} \beta_1 = \sup\{\eta | p_{\pi_+} = \eta p_{\pi_-} + (1-\eta)K, K \in \mathcal{C}\}, \\ \beta_2 = \sup\{\eta' | p_{\pi_-} = \eta' p_{\pi_+} + (1-\eta')K', K' \in \mathcal{C}\}, \end{cases} \tag{12}$$

*where $\mathcal{C}$ is the set containing all possible probability distributions. Thus, $\alpha$ is identifiable by*

$$\alpha = (1-\beta_1)/(1+\beta_1) \quad or \quad \alpha = 2\beta_2/(1+\beta_2). \tag{13}$$

*Proof.* We first prove the first line of (12). Since $p_{\pi_E}$ is irreducible w.r.t. $p_{\pi_{NE}}$, such that there exists no decomposition $p_{\pi_E} = (1-\eta)Q + \eta p_{\pi_{NE}}$ for any $\eta \in (0,1]$ and distribution $Q$, we have $p_{\pi_E}$ is also irreducible w.r.t. $p_{\pi_-}$, such that $p_{\pi_E}$ can not be rewritten as a mixture $p_{\pi_E} = (1-\eta')Q' + \eta' p_{\pi_-}$ for any distribution $Q'$ and $\eta' \in (0,1]$. Then, according to the relationship (5) of the main paper and Proposition 5 of [16], we can show $\beta_1$ is identifiable and $\beta_1 = \sup\{\eta | p_{\pi_+} = \eta p_{\pi_-} + (1-\eta)K, K \in \mathcal{C}\}$. We can prove the second line of (12) by a similar argument. Then, combined with $\beta_1 = \frac{1-\alpha}{1+\alpha}$ and $\beta_2 = \frac{\alpha}{2-\alpha}$ in (5) of the main paper, we can obtain identifiable $\alpha$ estimation in (13). $\qquad\square$

**Theorem 3.** *Let $\mathcal{W}$ be a parameter space for training the discriminator and $D_\mathcal{W} = \{D_w \mid w \in \mathcal{W}\}$ be the hypothesis space. Assume the functions $|\log D_w(s,a)| \leq B$ and $|\log(1 - D_w(s,a))| \leq B$ are upper-bounded for any state-action pair $(s,a) \in \mathcal{S} \times \mathcal{A}$ and $w \in \mathcal{W}$. Further assume both the functions $\log D_w(s,a)$ and $\log(1 - D_w(s,a))$ are L-Lipschitz continuous in the state-action space. For a fixed policy $\pi_\theta$, let $\Gamma^\theta = \{\tau_i^\theta\}_{i=1}^{n_\theta}$ be trajectories generated from $\pi_\theta$. Then, for any $\delta \in (0,1)$, with probability at least $1 - \delta$, we have*

$$V(\pi_\theta, D_w^*) - V(\pi_\theta, \widehat{D}_w) \leq 4L(1 + \alpha)\mathcal{R}_{n_+}(D_\mathcal{W}) + 4L(1 - \alpha)\mathcal{R}_{n_-}(D_\mathcal{W}))$$
$$+ 8L\mathcal{R}_{n_\theta}(D_\mathcal{W}) + C(\delta)\left(\frac{1}{\sqrt{n_\theta}} + \frac{1}{\sqrt{n_+}} + \frac{1}{\sqrt{n_-}}\right),$$

*where $\widehat{D}_w = \arg\max_{w \in \mathcal{W}} \widehat{V}(\pi_\theta, D_w)$ and $D_w^* = \arg\max_{w \in \mathcal{W}} V(\pi_\theta, D_w)$. The constants $n_+$ and $n_-$ are the number of trajectories in $\Gamma^+$ and $\Gamma^-$. We define $C(\delta) = 4B\sqrt{\log(6/\delta)}$. The empirical Radamacher complexities [40] on datasets $\Gamma^+$, $\Gamma^-$, and $\Gamma^\theta$ are denoted by $\mathcal{R}_{n_+}$, $\mathcal{R}_{n_-}$, and $\mathcal{R}_{n_\theta}$.*

*Proof.* Since $\widehat{D}_w$ and $D_w^*$ are the maximizer of the objective functions $\widehat{V}(\pi_\theta, D_w)$ and $V(\pi_\theta, D_w)$, respectively.

$$V(\pi_\theta, D_w^*) - V(\pi_\theta, \widehat{D}_w) = V(\pi_\theta, D_w^*) - \widehat{V}(\pi_\theta, D_w^*) + \widehat{V}(\pi_\theta, D_w^*) - \widehat{V}(\pi_\theta, \widehat{D}_w) \qquad (14)$$
$$+ \widehat{V}(\pi_\theta, \widehat{D}_w) - V(\pi_\theta, \widehat{D}_w)$$
$$\leq V(\pi_\theta, D_w^*) - \widehat{V}(\pi_\theta, D_w^*) + \widehat{V}(\pi_\theta, \widehat{D}_w) - V(\pi_\theta, \widehat{D}_w)$$
$$\leq 2\sup_{w \in \mathcal{W}}|V(\pi_\theta, D_w) - \widehat{V}(\pi_\theta, D_w)|, \qquad (15)$$

where the inequality is due to the optimality of $\widehat{D}_w$. The according to the definition of $V(\pi_\theta, D_w)$ and $\widehat{V}(\pi_\theta, D_w)$,

$$V(\pi_\theta, D_w) = 2\alpha\mathbb{E}_{(s,a)\sim p_{\pi_\theta}}[\log D_w(s,a)] + (1 - \alpha)\mathbb{E}_{(s,a)\sim p_{\pi_-}}[\log(D_w(s,a))] \qquad (16)$$
$$+ (1 + \alpha)\mathbb{E}_{(s,a)\sim p_{\pi_+}}[\log(1 - D_w(s,a))]$$

and

$$\widehat{V}(\pi_\theta, D_w) = 2\alpha\mathbb{E}_{(s,a)\sim \widehat{p}_{\pi_\theta}}[\log D_w(s,a)] + (1 - \alpha)\mathbb{E}_{(s,a)\sim \widehat{p}_{\pi_-}}[\log(D_w(s,a))] \qquad (17)$$
$$+ (1 + \alpha)\mathbb{E}_{(s,a)\sim \widehat{p}_{\pi_+}}[\log(1 - D_w(s,a))],$$

where $\widehat{p}_{\pi_\theta}(s,a) = \frac{1}{n_\theta}\sum_{i=1}^{n_\theta}\widehat{p}_i^\theta(s,a)$ and $\widehat{p}_i^\theta(s,a) = (1 - \gamma)\sum_{t=0}^{\infty}\gamma^t\mathbb{1}[(s_{t,i}, a_{t,i}) = (s,a)]$ is the empirical occupancy measure for the trajectory $\tau_i$ sampled from $\pi_\theta$The empirical occupancy measure is unbiased w.r.t. $\pi_\theta$ such that $\mathbb{E}[\widehat{p}_i^\theta(s,a)] = p_{\pi_\theta}(s,a)$ . Then, we can decompose the R.H.S. of (15) as

$$\sup_{w \in \mathcal{W}}|V(\pi_\theta, D_w) - \widehat{V}(\pi_\theta, D_w)| = \texttt{term (a)} + \texttt{term (b)} + \texttt{term (c)}.$$

In above, the first term

$$\texttt{term (a)} = 2\alpha\left|\mathbb{E}_{(s,a)\sim\widehat{p}_{\pi_\theta}}[\log D_w(s,a)] - \mathbb{E}_{(s,a)\sim p_{\pi_\theta}}[\log D_w(s,a)]\right|$$
$$= 2\alpha\left|\frac{1}{n_\theta}\sum_{i=1}^{n_\theta}\mathbb{E}_{(s,a)\sim\widehat{p}_i^\theta}[\log D_w(s,a)] - \mathbb{E}_{(s,a)\sim p_{\pi_\theta}}[\log D_w(s,a)]\right|$$

measures the generalization gap between $\widehat{p}_{\pi_\theta}$ and $p_{\pi_\theta}$. The second term

$$\texttt{term (b)} = (1 - \alpha)\left|\frac{1}{n_-}\sum_{i=1}^{n_-}\mathbb{E}_{(s,a)\sim\widehat{p}_i^-}[\log D_w(s,a)] - \mathbb{E}_{(s,a)\sim p_{\pi_-}}[\log D_w(s,a)]\right|$$

measures the generalization gap between $\widehat{p}_{\pi_-}$ and $p_{\pi_-}$ and the last term

$$\texttt{term (c)} = (1 + \alpha)\left|\frac{1}{n_+}\sum_{i=1}^{n_-}\mathbb{E}_{(s,a)\sim\widehat{p}_i^+}[\log(1 - D_w(s,a))] - \mathbb{E}_{(s,a)\sim p_{\pi_+}}[\log(1 - D_w(s,a))]\right|$$

measures the gap between $\widehat{p}_{\pi_+}$ and $p_{\pi_+}$.

Under the condition that $|\log D_w(s, a)| \leq B$ for any $(s, a) \in \mathcal{S} \times \mathcal{A}$ and $w \in \mathcal{W}$, the standard analysis generalization analysis with Rademacher complexity shows (e.g. Theorem 26.5 of [47]), for all $w \in \mathcal{W}$, we have

$$\texttt{term (a)} \leq 8\alpha \mathcal{R}_{n_\theta}(\log \circ D_{\mathcal{W}}) + 4\alpha B \sqrt{\frac{2 \ln(2/\delta')}{n_\theta}},$$

with probability at least $1 - \delta'$, where $\mathcal{R}_{n_\theta}$ is the empirical Rademacher complexity. Then, according to the Talagrand's contraction inequality (Lemma 26.9 of [47]), we have

$$\mathcal{R}_{n_\theta}(\log \circ D_{\mathcal{W}}) \leq L\mathcal{R}_{n_\theta}(D_{\mathcal{W}}),$$

Then, we obtain

$$\texttt{term (a)} \leq 8\alpha L\mathcal{R}_{n_\theta}(D_{\mathcal{W}}) + 4\alpha B \sqrt{\frac{2 \ln(2/\delta')}{n_\theta}}. \tag{18}$$

Since $|\log(1 - D_w)| \leq B$ and $\log(1 - D_w)$ is $L$-Lipschitz continuous, the a similar arguments ensures,

$$\texttt{term (b)} \leq 4(1 - \alpha)L\mathcal{R}_{n_-}(D_{\mathcal{W}}) + 2(1 - \alpha)B \sqrt{\frac{2 \ln(2/\delta')}{n_-}}. \tag{19}$$

and

$$\texttt{term (c)} \leq 4(1 + \alpha)\mathcal{R}_{n_+}(D_{\mathcal{W}}) + 2(1 + \alpha)B \sqrt{\frac{2 \ln(2/\delta')}{n_+}}, \tag{20}$$

with probability at least $1 - \delta'$. Let $\delta' = \delta/3$, we complete the proof by combining (18), (19), and (20) with (15).

$\square$

## B  Full Results of $\alpha$ Estimations

In the main body of the paper, we only reported the effect of overestimation and underestimation on HalfCheetah environment due to the space limitation. Here we provide the full results on four environments in our experiments, as shown in Figure 5. The experimental results demonstrated the same conclusion as in the main body of the paper, that the underestimation method is better than the overestimation one.

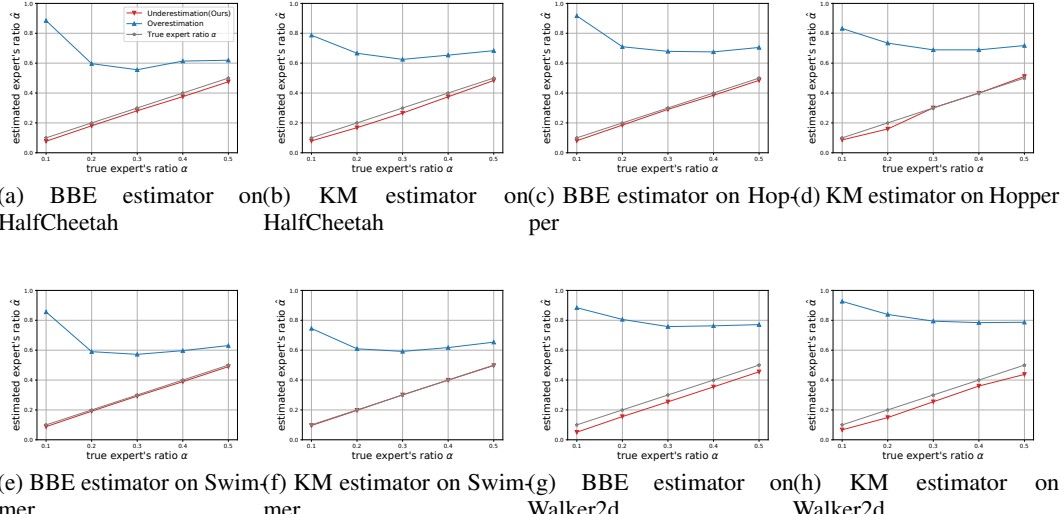

(a) BBE estimator on HalfCheetah    (b) KM estimator on HalfCheetah    (c) BBE estimator on Hopper    (d) KM estimator on Hopper

(e) BBE estimator on Swimmer    (f) KM estimator on Swimmer    (g) BBE estimator on Walker2d    (h) KM estimator on Walker2d

Figure 5: The comparisons of overestimated ($\frac{2\beta}{1+\beta}$) and underestimated ($\frac{1-\beta}{1+\beta}$) methods on various tasks with BBE and KM estimators.

