# OpenReview forum: "Imitation Learning from Vague Feedback"
_NeurIPS.cc/2023/Conference — NeurIPS 2023 poster_

### Official Review · Reviewer_CNdS · 2023-06-16

**Soundness:** 2 fair
**Presentation:** 2 fair
**Contribution:** 4 excellent
**Rating:** 6
**Confidence:** 3

**Summary:**

Edit: Changed the score from a 3 to a 6 due to the discussion below.

The paper focuses on tackling the challenge of Imitation Learning from imperfect demonstrations that come with vague pairwise comparisons. The authors make the key assumption that a competent annotator is available, but that they become uncertain when presented with either two expert or two non-expert samples, randomly assigning them as "good" or "bad" examples. To address this problem, they introduce a novel learning paradigm called Vaguely Pairwise Imitation Learning. By modeling the demonstration pool as a mixture of expert and non-expert data, the authors show that it is possible to recover the expert policy distribution when the proportion of expert data is known. Furthermore, they propose a method for estimating the mixture proportion in cases where this expert ratio is unknown.

To effectively leverage the recovered expert policy distribution, the authors integrate it with Generative Adversarial Imitation Learning (GAIL), resulting in end-to-end algorithms designed for scenarios with known and unknown expert data proportions. The performance of these algorithms is evaluated through experiments conducted on various Mujoco tasks. The results showcase that the proposed methods consistently outperform standard and preference-based imitation learning approaches for expert ratios of up to $0.5$.

**Strengths:**

- Novel Approach for Vaguely Pairwise Imitation Learning: The paper proposes a novel approach to estimate precise expert occupancy measures from essentially imperfect preferences. This addresses the challenge of working with imperfect data and provides a valuable new approach for imitation learning tasks.
- Sound Theory and Proofs: The paper demonstrates a solid theoretical foundation, supported by proofs in the appendix. The proposed algorithm is very closely related to the algorithm, using few additional assumptions or relaxations.
- Strong Experimental Performance: The experimental evaluation of the proposed method showcases its superior performance compared to a large number of standard and preference-based imitation learning methods. The results presented in the paper demonstrate the efficacy and practical relevance of the approach across various Mujoco environments.

Overall, the paper exhibits originality in its approach, a solid theoretical foundation for the presented algorithm, and significance in its experimental results. These strengths contribute to the advancement of the field of Imitation Learning and may offer valuable insights for both researchers and practitioners.

**Weaknesses:**

The following weaknesses are listed in order of severity.

- Suspected Issue with Overestimation: The values of the overestimation Figure 3a, b, and Appendix Figure 1a-h, are very close to 1 - the true ratio. Additionally, the matrix plots in Figure 3c, d appear as negative transposes of each other. This raises suspicion and suggests a potential issue with the code implementation.
- Inconsistent Experimental Results in Table 1: The experimental results presented in Table 1 exhibit inconsistencies. The HalfCheetah and Hopper experiments seem to use results from Swimmer and Walker2d for all methods, except for Behavioral Cloning and GAIL. Consequently, there are no results provided specifically for these tasks in the current state of the submission, which reduces the completeness of the evaluation.
- Evaluation of GAIL_expert and Confidence Intervals: The evaluation of GAIL_expert in Table 1 is performed separately for different alpha values, despite its independence from alpha. Moreover, while GAIL_expert is presented as an upper bound for performance, the presented methods outperform it in several instances. This may indicate that the number of random seeds used (5) is insufficient, especially considering the relatively large confidence intervals.
- Unclear Advantages of COMPILER-E: It is unclear why COMPILER-E would outperform COMPILER. Exploring potential reasons, for example that COMPILER-E may remove "bad" expert demonstrations through an implicit de-noising procedure in its estimation of the expert ratio could provide clarification and improve the understanding of its advantages.
- Limited Experiment Scope: The experiments conducted in the paper are limited to Mujoco environments. Including experiments on other environments, such as Meta-world, would enhance the comprehensiveness of the evaluation.
- Code Unavailability: The paper does not provide the code, which can hinder reproducibility and limit further exploration of the proposed methods.
- Missing Hyperparameters: The paper lacks information about the hyperparameters used for some of the baseline methods. For example, for D-Rex the number of noise levels utilized to estimate the distributions during training is missing.
- Proof Derivations Lack Readability: The derivations for the proofs are presented in a compact manner, making them difficult to follow. Providing additional steps and explanations would greatly improve the readability of the proofs.
- Typos and Style:
    - Line 133 references Section 6.4, which does not exist in the paper.
    - The layout of the caption for Figure 3 makes it challenging to read and comprehend the information provided.
    - Figure 2 could benefit from a descriptive caption to aid understanding and provide context for the information presented.
    - There are instances of redundancy and language issues throughout the paper, such as the information in Line 163, which was already mentioned in the previous paragraph and e.g., the over-use of "the" in the first sentence of the description of Figure 1. Addressing these issues would improve the overall clarity and readability of the paper.
    - Factorization in Equation 3: In Equation 3, the factor $2(1-\alpha)$ is factored out only once, even though it appears twice in the equation. Ensuring consistent notation would enhance the equation's clarity.

Overall, the suspected problems with the overestimation method for the expert ratio, and the missing results for $2$ out of $4$ tasks are a clear reason to reject the paper. Addressing these (and other) issues during the rebuttal period would significantly increase the quality of the submission.

**Questions:**

- Importance of Equation 8: Equation 8 rephrases the GAIL loss in terms of the estimated positive and negative data distributions to mitigate overfitting. How crucial is this rephrasing for the overall performance of the approach? Conducting an ablation study to assess the impact of this choice would provide valuable insights and clarify its significance.
- Alternatives to Algorithm 1: Algorithm 1 includes the design choices of iteratively training a discriminator to approximately separate expert from non-expert data. What would happen if a mixture proportion estimator were directly applied to \Gamma^+ and \Gamma^- instead? Explaining the reasoning behind the chosen approach and comparing it to alternative strategies would enrich the discussion and highlight the advantages of the proposed algorithm.
- Intuition Behind the Step from Equation 2 to Equation 3: Equations 2 and 3 explain the relations between the (non)-expert the pairwise dataset’s occupancy measures. It would be helpful to gain some intuition on what the transition from e.g., $p_{\pi_+}$ to $p_{\pi_E}$ actually means. A deeper explanation or examples illustrating this step would make this easier to understand.
- Clarification on Expert Ratio Estimation: Lines 96-98 state that "[…] [5] also used the risk rewriting technique to identify the expert demonstrations, but their algorithm does not require estimation of the expert ratio as the confidence score is given." For the presented approach, the expert ratio is also given in some of the experiments. How does the approach differ from this particular related work in this case?

**Limitations:**

The limitations or potential negative societal impact of the work are currently not addressed. It should be acknowledged that the method assumes a perfect annotator who randomly assigns samples when they come from the same distribution. However, in real-world scenarios, annotators may possess biases or preferences towards certain types of solutions, which are not accounted for in the proposed approach. Addressing and discussing this limitation would strengthen the submission.

In terms of ethical considerations, aspects such as fairness, transparency, and bias in the training and deployment of imitation learning models is crucial to ensure responsible and ethical AI development. It could be beneficial for the authors to include a section that highlights these potential ethical implications.

---

> ### Author Rebuttal · Authors · 2023-08-10
>
> We are grateful for the comprehensive feedback and deep dive into our work. We value the acknowledgment of our work's novelty, theoretical rigor, and experimental robustness. We have provided the correct results in the general response's PDF file and have shared the code and demonstrations from our experiments for reproducibility. We hope this assuages any concerns. For brevity, we will focus our response on addressing technical queries.
> ### Q1:
> **Query:** "Suspected Issue with Overestimation"
>
> **Response:** We appreciate your meticulous examination of our empirical results. The phenomenon observed in Figure 3a,b is attributed to the inherent limitations of the overestimation method, particularly when $\alpha$ is less than 0.5. During overestimation, as guided by Eq.(7), we discern the maximum proportion of $p_{\pi_+}$ within $p_{\pi_-}$ as $\beta$ , subsequently computing $\alpha = \frac{2\beta}{1+\beta}$ . In scenarios where $\alpha$ is diminutive, the estimation of $\beta$ can be notably unreliable. This discrepancy is further elucidated with a detailed example: For example, in the case when $\alpha = 0.1$, we have $p_{\pi_+} \approx 0. 2p_{\pi_E} + 0.8p_{\pi_{NE}} $ and $p_{\pi_{-}} \approx 0. 01p_{\pi_E} + 0.99p_{\pi_{NE}}$. Since the proportion of $\pi_E$ in $p_{\pi_+}$ is larger than that in $p_{\pi_-}$, we expected the estimator to return the $\beta \approx 0. 01p_{\pi_E} / 0. 2p_{\pi_E} = 0.05$, which finally lead to the right estimation $\alpha \approx 0.1$. However, in practice, the mixture proportion will be highly affected by the data from $\pi_{NE}$. Since both $p_{\pi_+}$ and $p_{\pi_-}$ contain a large proportion of $p_{\pi_{NE}}$, the MPE estimator will take them as almost the same distribution and return the value $\beta$ close to 1, thus leading to $\alpha$ close to 1, which is in accordance with the results in Figure 3(a) and Figure 3(b).
>
> Regarding Figure 3(c and d), we apologize for the confusion. The heatmap values essentially represent theoretical computations rather than algorithmic outputs. These figures aim to elucidate the relationship between actual $\alpha$ values and the estimated $ \hat{\alpha} $. We will clarify this in our subsequent revision.
> ### Q2:
> **Query:** "Inconsistent Experimental Results in Table 1"
>
> **Response:** We are grateful for highlighting this oversight. The corrected results have been furnished in the general response's pdf. Furthermore, for the sake of reproducibility, we have made available the codes and all experimental demonstrations. We sincerely hope that these corrective actions address the concerns, and we pledge to rectify these errors in our revised version.
> ### Q3:
> **Query:** "The evaluation of GAIL_expert in Table 1 is performed separately for different alpha values... the number of random seeds used (5) is insufficient"
>
> **Response:** It's crucial to note that GAIL_expert exclusively utilizes expert data extracted from mixed data. Consequently, with a diminishing $\alpha$ , the volume of expert data may reduce, necessitating separate evaluations for distinct $\alpha$ values in Table 1. Besides, five seeds is a customary selection in the IL domain [1,2,3]. Results are deemed acceptable as long as the method doesn't considerably surpass the expert's performance.
> ### Q4:
> **Query:** "It is unclear why COMPILER-E would outperform COMPILER."
>
> **Response:** We have included t-tests comparing COMPILER and COMPILER-E in the general response's pdf. It's evident that COMPILER-E's superiority over COMPILER is restricted to specific tasks and $\alpha$ values. In a majority of cases, COMPILER outshines COMPILER-E. Thus, instances where COMPILER-E outperforms COMPILER are mere anomalies.
> ### Q5:
> **Query:** "Limited Experiment Scope: The experiments conducted in the paper are limited to Mujoco environments. Including experiments on other environments, such as Meta-world, would enhance the comprehensiveness of the evaluation."
>
> **Response:** We concur on the importance of method verification across diverse tasks. In the present submission stage, our primary objective is to demonstrate our algorithm's prowess in handling mixed demonstrations from both experts and non-experts within VPIL problems. Notably, Meta-world, driven by MuJoCo's physics engine, primarily serves Meta-Learning scenarios. Considering VPIL problems, we have showcased the efficacy of COMPILER and COMPILER-E across various continuous control tasks.
> ### Q6:
> **Query:** "How crucial is this rephrasing for the overall performance of the approach?"
>
> **Response:** The reframing of the GAIL loss in Eq. 8 is pivotal for our algorithm's success in VPIL problems. In the conventional IL paradigm, policies can be optimized by minimizing the original GAIL loss based on the expert policy $p_{\pi_{E}}$ . However, in VPIL problems, the expert policy remains elusive due to the inherent ambiguity of feedback, thereby complicating the evaluation of the original GAIL loss. The equivalence between the original GAIL loss and the rephrased version (Eq. 8) allows us to evaluate the original loss using the vague datasets $\Gamma^+$ and $\Gamma^-$ .
> ### Q7:
> **Query:** "It would be helpful to gain some intuition on what the transition from e.g., $p_{\pi_+}$ to $p_{\pi_E}$ actually means."
>
> **Response:** Eq. 2 can be interpreted as a linear system with two variables. Within our framework, both $p_{\pi_E}$ and $p_{\pi_{NE}}$ are the elusive variables we aim to decipher, while all other metrics, including p_{\pi_+} and p_{\pi_-} , are empirically estimable. Thus, the transition from Eq. 2 to Eq. 3 mirrors the linear system resolution process. Through this mechanism, we can represent $p_{\pi_E}$ and $p_{\pi_{NE}}$ as linear combinations of $p_{\pi_+}$ and $p_{\pi_-}$ .
>
> ## Reference
> [1] Learning to Weight Imperfect Demonstrations. ICML 2021.
>
> [2] Learning from Suboptimal Demonstration via Self-Supervised Reward Regression. CoRL 2020.
>
> [3] Imitation Learning from Imperfect Demonstration. ICML 2019.

---

> > ### Comment · Reviewer_CNdS · 2023-08-11
> >
> > I want to thank the authors for the detailed answer and for clearing up most of my questions.
> >
> > > [...] Regarding Figure 3(c and d), we apologize for the confusion. The heatmap values essentially represent theoretical computations rather than algorithmic outputs. These figures aim to elucidate the relationship between actual $\alpha$
> >  values and the estimated $\hat{\alpha$}. We will clarify this in our subsequent revision.
> >
> > In this case, I am wondering why the overestimation would be used at all. From the provided figures in the paper, the used approach seems to accurately represent the desired quantity, while the overestimation variant essentially always returns "1-quantity". Though I deeply appreciate the additional explanation, I feel like I still do not fully understand this "1-quantity"-relationship, and consequently that the heatmaps are essentially inverse transposes of each other. I kindly ask the authors to further clarify potential uses of the overestimation variant and to provide some more details on the "inverse" relationship between it and the method used throughout the rest of the paper.
> >
> > > Query: "Inconsistent Experimental Results in Table 1"
> > > Response: We are grateful for highlighting this oversight. The corrected results have been furnished in the general response's pdf. Furthermore, for the sake of reproducibility, we have made available the codes and all experimental demonstrations. We sincerely hope that these corrective actions address the concerns, and we pledge to rectify these errors in our revised version.
> >
> > I believe that the inclusion of the full results and especially the source code significantly improves the quality of the submission and want to thank the authors for providing them.
> >
> > > Q3: Query: "The evaluation of GAIL_expert in Table 1 is performed separately for different alpha values... the number of random seeds used (5) is insufficient" [...]
> > > Response: It's crucial to note that GAIL_expert exclusively utilizes expert data extracted from mixed data. Consequently, with a diminishing $\alpha$ , the volume of expert data may reduce, necessitating separate evaluations for distinct $\alpha$ values in Table 1. Besides, five seeds is a customary selection in the IL domain [1,2,3]. Results are deemed acceptable as long as the method doesn't considerably surpass the expert's performance.
> >
> > Does this mean that the differen GAIL_expert runs (for different values of $\alpha$) train on different subsets of the expert demonstrations? If so, why? In my opinion, using more than 5 seeds would still be beneficial here regardless of precedents set by related work, as dealing with vague data likely leads to less reliable convergences and thus scores. I think this can be seen in the large and largely overlapping standard deviations of the results. If the authors disagree, I would be interested in hearing why.
> >
> >
> > Overall, the authors addressed most of my questions and concerns, and I would be glad to raise my score if these last remaining points are cleared up.

---

> > > ### Author Response · Authors · 2023-08-14
> > >
> > > We sincerely appreciate the time and effort the reviewer has taken to provide detailed feedback. Herein, we address the queries raised.
> > >
> > > ---
> > >
> > > **Q1:** “In this case, I am wondering why the overestimation would be used at all. From the provided figures in the paper, the used approach seems to accurately represent the desired quantity, while the overestimation variant essentially always returns "1-quantity". Though I deeply appreciate the additional explanation, I feel like I still do not fully understand this "1-quantity"-relationship, and consequently that the heatmaps are essentially inverse transposes of each other. I kindly ask the authors to further clarify potential uses of the overestimation variant and to provide some more details on the "inverse" relationship between it and the method used throughout the rest of the paper.”
> > >
> > > **A1:** Theoretically, when both datasets are infinitely large such that $p_{\pi_+}$ and $p_{\pi_-}$ can be accurately estimated, both over and underestimation methods would achieve high precision. However, as previously discussed in Response of Q1 in rebuttal, the overestimation method actually employs $p_{\pi_E}$ to calculate the expert ratio (for instance, at $\alpha = 0.1$, we compute the expert ratio using $\beta \approx 0.01 p_{\pi_E}/0.2 p_{\pi_E} =0.05$). When $ \alpha < 0.5$ (indicating a predominance of non-experts in the dataset), the overestimation method becomes significantly influenced by non-expert data. We aimed to empirically verify this observation, and our experimental results indeed support this conclusion.
> > >
> > > To provide more clarity, we plotted the estimated values from both overestimation and underestimation methods across each training step. These plots, which illustrate the training process of expert ratio estimation using the BBE algorithm, can be found at the **anonymized GitHub link** provided in the global response (specifically under **./estimation_figures/{env}.BBE.alpha_{alpha_value}.png**, encompassing a total of 20 figures). The KM results will be added to the link once finished. These plots depict the evolution of estimated expert ratios $\hat \alpha$ during each training epoch. As depicted in the figures, the relationship between overestimation and underestimation is not strictly "1-quantity" (though quite similar) during the distinct stages of the training.
> > >
> > > We recognize the confusion in the paper and will take measures to clarify this by adding the records and updating the figures accordingly. It remains our conclusion that, especially for VPIL problems with low $\alpha$, the overestimation approach is less optimal than the underestimation one.
> > >
> > > For the heatmap concern, we interpret the “inverse” concern as regarding the “transposition” of matrices (we are glad to hear the reviewer’s comments if we misunderstand it). **Figure 3(c-d)** offers a **theoretical insight**, depicting the effects of imprecise estimates in terms of over- or under-throwing from the datasets. Specifically, from Eq. (2), we want to throw $(1-\hat\alpha)^2$ ratio of non-expert data in $\Gamma^+$ and $\hat{\alpha}^2$ ratio of expert data in $\Gamma^-$. To investigate the over- and under-throwing data, we calculate the difference matrices $(1-\hat\alpha)^2-(1-\alpha)^2$ in $\Gamma^+$ and $\hat{\alpha}^2-\alpha^2$ in $\Gamma^-$, as shown in Figure 3(c-d). The two matrices are only numerically transposed, but express different meanings. Conversely, **Figure 3(a-b)** is an **empirical study**, directly illustrating the effects of over and underestimation. Thus, Figures 3(c-d) and 3(a-b) offer complementary perspectives to comprehensively evaluate the performance of our algorithm.

---

> > > > ### Author Response · Authors · 2023-08-14
> > > >
> > > >
> > > > **Q2:** “I believe that the inclusion of the full results and especially the source code significantly improves the quality of the submission and want to thank the authors for providing them.”
> > > >
> > > > **A2:** We genuinely thank the reviewer for recognizing the updated results and source code. Committing to transparency and robustness, we will ensure that the full results are included in our revision, and the source code is released post-review. We believe that such actions fortify the integrity of our submission.
> > > >
> > > > ---
> > > >
> > > > **Q3:** “Does this mean that the differen GAIL_expert runs (for different values of $\alpha$) train on different subsets of the expert demonstrations? If so, why? In my opinion, using more than 5 seeds would still be beneficial here regardless of precedents set by related work, as dealing with vague data likely leads to less reliable convergences and thus scores. I think this can be seen in the large and largely overlapping standard deviations of the results. If the authors disagree, I would be interested in hearing why.”
> > > >
> > > > **A3:** Prior to the current submission, we did run experiments where GAIL_expert was trained using consistent expert demonstrations. However, in practice, distinct $\alpha$ values arise due to variability in VPIL problems, leading to **divergent $\Gamma^+$ and $\Gamma^-$** datasets by annotators **for all other methods**. To ensure experimental fairness, we opted to sample **varying expert demonstration subsets** under different $\alpha$ values **for GAIL_expert**.
> > > >
> > > > We agree with the reviewer's comments about the potential advantages of more trials for improved reliability. Yet, given our extensive experimentation across 11 methods and 20 VPIL tasks, we found that 5 trials—consistent with past imitation learning studies—sufficiently highlight significant methodological differences, which is further supported by our inclusion of a 95% t-test comparing COMPILER and COMPILER-E. We are open to conducting more trials in our revision, but anticipate that our primary conclusions will stand firm.
> > > >
> > > > ---
> > > >
> > > > We are grateful for the reviewer's constructive feedback and will endeavor to address these concerns comprehensively in our revised manuscript. We hope our response alleviates your concerns.

---

> > > > > ### Comment · Reviewer_CNdS · 2023-08-15
> > > > >
> > > > > I want to sincerely thank the reviewers for clearing up my remaining concerns. I will raise my score accordingly.

---

> > > > > > ### Author Response · Authors · 2023-08-15
> > > > > > **Thank you**
> > > > > >
> > > > > > We appreciate the reviewer for valuable feedback and for raising the score!

---

### Official Review · Reviewer_DWbY · 2023-06-30

**Soundness:** 2 fair
**Presentation:** 2 fair
**Contribution:** 2 fair
**Rating:** 5
**Confidence:** 2

**Summary:**

This paper falls into imitation learning with human preference. Specifically, this paper deals with a more challenging setting, where the annotator information is wrong, i.e., some data collected by non-experts could be labeled as positive demonstrations, and some data collected by experts could be labeled as negative demonstrations.
This paper proposed two methods to solve this challenge, one with a known ratio of real demonstration data, and one with an unknown ratio.
The proposed approach is theoretically justified, and evaluated in the mujoco domain.

**Strengths:**

The proposed method is theoretically justified.

**Weaknesses:**

The experiment results in rather confusing, e.g., in Table 1, for the HalfCheetah environment, apparently, the return of COMPILER is not the largest, but it is with the bold font, and the return of GAIL if significantly larger than the proposed approach.

**Questions:**

If the authors could show that the proposed method can work well in a more challenging task, such as humanoid running, this paper could be greatly improved.

**Limitations:**

This paper has not discussed the limitations in the main body of the submission.

---

> ### Author Rebuttal · Authors · 2023-08-10
>
> We thank the reviewer for highlighting the inconsistencies in Table 1. An updated version of Table 1 has been made available in the general response's PDF file. We have also ensured reproducibility by providing the code and demonstrations from our experiments via anonymous links. We are committed to addressing these concerns in our revised manuscript. Below, we delve into a discussion on potential further experiments.
>
> ### Q1:
> **Query:** "If the authors could show that the proposed method can work well in a more challenging task, such as humanoid running, this paper could be greatly improved."
>
> **Response:** We concur that validating our proposed method on more demanding tasks would be beneficial. We have earmarked such tasks for the project's scaling-up phase. In the current submission, our objective was to showcase the efficacy of handling mixed demonstrations from both experts and non-experts in VPIL scenarios. While we have incorporated multiple continuous control tasks analogous to humanoid running in MuJoCo, the proposed method has displayed commendable results in VPIL scenarios. However, the sheer volume of required experiments, given the numerous baselines and our methods across various \( $\alpha$ \) levels in a single environment (and the need for multiple trials for credibility), precluded the inclusion of these tasks in the rebuttal timeframe. Nonetheless, we are keen on validating our methods on more challenging tasks in the future.

---

> > ### Comment · Reviewer_DWbY · 2023-08-12
> > **Response**
> >
> > Thank you for taking the time to answer the questions. Most of my concerns have been cleared, and I will raise the score.

---

> > > ### Author Response · Authors · 2023-08-15
> > > **Thank you**
> > >
> > > We appreciate the reviewer for valuable feedback and for raising the score!

---

### Official Review · Reviewer_g4mG · 2023-07-05

**Soundness:** 3 good
**Presentation:** 4 excellent
**Contribution:** 4 excellent
**Rating:** 9
**Confidence:** 5

**Summary:**

The manuscript considers a popular learning framework, learning from human feedback/preference. In real-world applications, the feedback can be inaccurate and detrimental to the training process. The manuscript formulates the imitation learning (IL) from vague feedback process, in which the data annotator can only provide accurate preference when the paired demonstrations are significantly different. The authors analyze the problem for two different levels of difficulty, i.e., known or unknown expert proportion $\alpha$, and propose two algorithms based on risk rewriting and mixture proportion estimation to solve the two problems separately. The authors also conduct experiments and case studies with various IL tasks and state-of-the-art baselines to verify the effectiveness and generality of the algorithms.

**Strengths:**

### Originality
The manuscript introduces a realistic problem in feedback-based imitation learning, accounting for the potential noisiness in human feedback. The proposed scenario is distinguished by its realism and poses novel challenges to the reinforcement learning and imitation learning communities. Furthermore, the authors ingeniously develop a learning framework, which, when integrated with existing imitation learning techniques, effectively addresses the stated problem.

### Quality
The authors adeptly formulate the problem, lending credence to its practical significance. The learning framework presented is versatile, facilitating integration with various imitation learning (IL) and mixture proportion estimation (MPE) methods. Commendably, the manuscript offers rigorous theoretical analyses regarding the interaction between MPE and the Generative Adversarial Imitation Learning (GAIL) approach. Moreover, the experimental section is robust, featuring comparisons across an extensive array of tasks and benchmarks. This establishes the efficacy and generalization capabilities of the proposed algorithms.

### Clarity
The writing is lucid, and the claims advanced are substantiated effectively. The incorporation of a comprehensive collection of case studies furnishes empirical validation, thus bolstering the merits of this work. This meticulous detailing is, in my estimation, one of the strong suits of this manuscript.

### Significance
This manuscript addresses a salient issue within the reinforcement learning and imitation learning domains - learning from human feedback. Through an adept modeling of ambiguous feedback and the introduction of an efficacious learning framework, this manuscript renders important contributions to the field. These are all very important contributions to the community.

**Weaknesses:**

The text size in Table 1 is a little too small. Also, I recommend the authors to put the robustness results in Appendix D to the main manuscript. This is also an important case study.

**Questions:**

Is it possible to separate expert and non-expert data first, and then use T-REX to learn the policy?

**Limitations:**

Although the authors have illustrated VPIL with multiple annotators as the future work, I recommend the authors to discuss it as a limitation of this work with more details.

---

> ### Author Rebuttal · Authors · 2023-08-10
>
> We are truly grateful for the reviewer's encouraging comments and constructive feedback. We will address some of the technical inquiries below.
>
> ### Q1:
> **Query:** "I recommend the authors to put the robustness results in Appendix D to the main manuscript. This is also an important case study."
>
> **Response:** We acknowledge the importance of the robustness results when encountering the noise on $\alpha$. We will endeavor to integrate these results into the main manuscript during our revision, ensuring they receive the attention they merit.
>
> ### Q2:
> **Query:** "Is it possible to separate expert and non-expert data first, and then use T-REX to learn the policy?"
>
> **Response:** Your suggestion offers an intriguing perspective. However, conventional preference-based algorithms, including T-REX, do not yield satisfactory results in VPIL scenarios when applied in this manner. It is because that T-REX necessitates comprehensive ranking data, but post-separation, we only have binary datasets, which are informationally inferior to full-ranking datasets.

---

### Official Review · Reviewer_B2bu · 2023-07-09

**Soundness:** 3 good
**Presentation:** 2 fair
**Contribution:** 3 good
**Rating:** 5
**Confidence:** 3

**Summary:**

The paper formulates the problem of imitation learning from imperfect demonstrations with vague comparisons named Vaguely Pairwise Imitation Learning (VPIL) which contains both expert and non-expert demonstrations, and the data collector only provides vague pairwise information of demonstrations. The authors proposed two learning paradigms to solve VPIL. One with risk rewriting and mixture proportion estimations (MPE), to recover the expert distribution with the known expert ratio α and unknown one respectively. They evaluated the proposed method on simulate MuJoCo based tasks.

**Strengths:**

- The paper attempts to solve a pretty interesting problem.
- The experimental results demonstrates that the proposed method is able to outperform standard and preference-based IL methods.
- The paper presents good theoretical analysis of the problem and proposed method.
- Extensive comparison to baselines.


**Weaknesses:**

- The paper has some typos and grammatical errors. For ex. In line 333, "the" is written twice; In line 36, "off-the-shall" -> "off-the-shelf". I would highly recommend a senior author to proof-read the paper.
- Some parts of the paper are a little difficult to follow. For ex. Line 43-53.
- Also, the problem of vague feedback is a little unclear.
- The paper evaluates the method on 4 different MuJoCo tasks. However, problem is motivated from real-world robotics/imitation based task. It would be good for the authors to at least discuss how there method would work in real-world. If possible they can perform some simple experiment on real-world task that would be great.
- There is only one small future extension mentioned in the paper. The authors are encouraged to at least mention few more and have breif discussion.

**Questions:**

Mentioned above in weakness

**Limitations:**

- I don't think authors have mentioned the limitations of the proposed method. They are encouraged to mention them in the paper.

---

> ### Author Rebuttal · Authors · 2023-08-10
>
> We're appreciative of the reviewer's feedback, especially the emphasis on enhancing clarity. We will address the technical and motivation-related queries below.
>
> ### Q1:
> **Query:** "Some parts of the paper are difficult to follow, e.g., Line 43-53 and the problem of vague feedback is unclear."
>
> **Response:** In lines 43-53, we described a real-world data collection scenario. Contrasting with Figure 1c, where the annotator provides a clear global comparison, in our problem, the annotator can only offer a vague comparison. This process is elucidated in Figure 2. We acknowledge the ambiguity and will enhance clarity regarding vague feedback in our revision.
>
> ### Q2:
> **Query:** "Discussion on how the method would work in the real-world."
>
> **Response:** In real-world applications, the data collection process illustrated in Figure 2 can be directly employed to curate \( $\Gamma^+$ \) and \($ \Gamma^- $\) datasets. This approach minimizes the annotator's burden as they only need to provide vague feedback for data pair differentiation. Subsequently, our algorithms (COMPILER for known \($ \alpha$ \) scenarios and COMPILER-E for unknown \( $\alpha$ \) cases) can be employed to derive the desired policy. We value this suggestion and will incorporate it into our revised conclusion.
>
> ### Q3:
> **Query:** "There is only one small future extension mentioned in the paper. The authors are encouraged to at least mention few more and have breif discussion.."
>
> **Response:** We concur that the exploration of the broader applicability of our work is valuable. In addition to the two potential extensions in our manuscript, we envisage other promising directions, such as:
>
> 1. **Active Learning Integration:** An active learning paradigm could enable the agent to judiciously query human annotators for clear feedback on demonstrations that would most refine the imitation model.
> 2. **Hierarchical Imitation Learning with Vague Feedback:** Implementing hierarchical models might offer a nuanced approach to assimilate vague feedback across varying task granularities.
> 3. **Integration with Other Feedback Mechanisms:** Exploring the synergy between the vague feedback mechanism and other feedback types, like reward shaping or advice-giving, could augment the learning process.
>
> We aim to incorporate these discussions into our revised manuscript, contingent on space availability.

---

### Official Review · Reviewer_zY91 · 2023-07-26

**Soundness:** 3 good
**Presentation:** 4 excellent
**Contribution:** 3 good
**Rating:** 6
**Confidence:** 3

**Summary:**

The paper studies the problem of Imitation Learning (IL) when the provided demonstrations come from experts and non-experts. The dataset may then not divided exactly into expert and non-expert subsets, but instead a data collection procedure may only vaguely separate the data. Thus, the authors pose the problem of Vague pairwise imitation learning, where the goal is finding a policy that approximates the expert policy that provided the expert demonstrations.
To tackle the problem, two cases are considered: the ratio of expert to non-expert demonstrations is known or unknown. In the first case the authors show how the occupancy measure (commonly used in IL) of the expert policy can be estimated, which then allows to solve the problem using standard IL methods. In the second case, the ratio is estimated using a lower bound.
The approach is evaluated in 4 different simulation setups and compared against numerous state-of-the-art IL methods. For various ratios of expert to non-expert data, the proposed method shows clear advantages over the baselines. Lastly, the evaluation provides a robustness analysis for the estimator, as well as empirical results motivating the use of the lower bound as opposed to an upper bound.


**Strengths:**

Overall, the paper is very well written and addresses an interesting and active research area.
This work addresses limitations of state-of-the-art IL algorithms when the dataset is a mixture of expert and non-expert demonstrations.
The proposed approach is - to the best of my knowledge - original  and the technical contribution is sound and provides theoretical results on the way. The simulation results are overall convincing and effectively showcase the benefits of the proposed method.

**Weaknesses:**

The structure of the paper could be improved by adding a clear problem statement and list the main contributions.
In the problem description the paper should add some more detail, in particular definitions of the expert and non-expert policies.
Some parts of the discussion of the results should be clarified.
Further, the paper does not have a discussion of its limitations and how restrictive the assumptions / model for the data collector are.
 Please find my detailed comments below.


**Questions:**

Questions

1.	Problem statement (Sect. 3.2): There is no definition of an expert and non-expert policy. How do they differ? Is an expert always consistent, i.e., do their demonstrations optimizes the same reward function (without any noise)? It would be good to define the policy types at least on a high level to obtain a well-posed problem.

2.	No clear statement of contribution: The introduction would benefit from a contributions section, highlighting the novelty of the work.

3.	Results: Line 292 “As α grows, […] COMPILER and COMPILER-E got the biggest performance boost and achieved the best performances.” This claim does not seem to fit the data reported in the table. On two of four tasks it is CAIL that achieves the best performance for the largest alpha=0.5. Furthermore, the relative increase between alpha 0.1 and 0.5 is only up to 5% for COMPILER, while CAIL increases the return by 50-100%. Either I misunderstand something, or the claim is wrong, please clarify.

4.	The paper should have limitations section, discussing how restrictive the model for the data collection is. Further, related to my first question, some discussion on the differences between the expert and non-expert policies would be required, and how accurate the assumed policies are.


Minor Comments

5.	What are the objectives in the maze example in Figure 1e? Only distance is mentioned, but with only one objective the task becomes rather trivial, even for non-experts

6.	End of problem statement: It would be good to conclude the section with a concise problem statement (in a theorem-style environment). Earlier it is mentioned that the goal is to learn a policy $\pi_{\theta}$ that mimics the expert policy – does that mean it minimizes the difference in the occupancy measure? It would help to make such an objective explicit.

7.	Lines 135-136  seem to be redundant w.r.t. to the paragraph before

8.	Line 160. What algorithm is referred to? Is (3) referring to equation 3? I cannot really follow the sentence, this should be clarified.

9.	Results: The problem statement defined alpha to be in (0,1], yet the experiments only consider alpha from .1 to .5. What happens for larger alpha – do the baselines catch up eventually? These results might not be as interesting as for the lower alpha reported, but this could at least be mentioned. Also it is peculiar in Fig 3 a and b that the over and underestimate already converge for alpha=.5 – is there a reason for that?


**Limitations:**

The paper does not have a limitations section.

---

> ### Author Rebuttal · Authors · 2023-08-10
>
> We express our gratitude to the reviewer for their positive remarks and in-depth engagement with our problem setting. We will now address the concerns raised.
> ### Q1:
> **Query:** "no definition of an expert and non-expert policy. How do they differ?"
>
> **Response:** Our approach adheres to the foundational definitions from the imitation learning from imperfect demonstrations community, where both the expert and non-expert operate within the same environment with a unified goal (the same reward function) [1,2,3]. Specifically, for our experiments, we trained converged RL agents as experts to produce expert demonstrations, while the unconverged agents were treated as non-experts. The performance metrics of both experts and non-experts can be found in Appendix B.
> ### Q2:
> **Query:** "No clear statement of contribution."
>
> **Response:** Our primary contributions are:
>
> 1. Introduction of the Vaguely Pairwise Imitation Learning (VPIL) problem. In VPIL, data annotators, when faced with the challenge of distinguishing between two trajectories, can only provide ambiguous feedback. The agent's objective is to derive an effective policy from such nebulous information.
> 2. We proposed methods to extract the expert policy distribution from a mix of expert and non-expert demonstrations using risk rewriting and mixture proportion estimation techniques.
> 3. Integration of our methodology with existing imitation learning algorithms, such as GAIL, resulted in two end-to-end solutions, COMPILER and COMPILER-E, for VPIL problems with and without \( $\alpha$ \). Additionally, we conducted rigorous mathematical analyses to ensure method convergence.
> 4. A comprehensive set of experiments were undertaken to validate our claims. The results showcased the superiority of our techniques over traditional and preference-based imitation learning methods, emphasizing their effectiveness in handling ambiguous feedback.
>
> We acknowledge the oversight of not explicitly listing these contributions in the original submission due to space constraints. We will ensure their inclusion in the revision.
> ### Q3:
> **Query:** "'COMPILER and COMPILER-E got the biggest performance boost and achieved the best performances.' This claim does not seem to match the data reported in the table."
>
> **Response:** We appreciate your attention to this matter. An oversight led to this erroneous statement, and we will rectify it in our revision. Our algorithm exhibits a diminishing performance boost as \( $\alpha$ \) increases. This is because even at \($ \alpha = 0.1$ \), our proposed method had already achieved expert-level performance, comparable to GAIL_expert, which is trained exclusively with expert data.
> ### Q4:
> **Query:** "The paper should have a limitations section"
>
> **Response:** We have briefly touched upon the limitations in lines 135-136 and 343-344 of our manuscript. Here, we indicated that our framework doesn't factor in noise introduced by potential attackers or annotator errors. Additionally, we only considered a single annotator for VPIL problems, whereas real-world scenarios might involve multiple annotators with distinct preferences. In response to your feedback, we will restructure and elaborate on these limitations in our revision.
> ### Q5:
> **Query:** "What are the objectives in the maze example in Figure 1e?"
>
> **Response:** The primary goal in this example was to demonstrate that even with a simplistic objective, like distance, annotators might struggle to provide a clear preference between two trajectory pairs. This serves as an illustrative motivation. For our experiments, we incorporated more intricate continuous control tasks to validate the efficacy of our methods.
> ### Q6:
> **Query:** "It is mentioned that the goal is to learn a policy $\pi_\theta$ that mimics the expert policy – does that mean it minimizes the difference in the occupancy measure?"
>
> **Response:** The aim of learning a policy $\pi_\theta$ to mimic the expert policy amidst mixed demonstrations is central to VPIL problems. However, the methodology we adopted, which involves minimizing the difference in the occupancy measures between the learner's and the expert's policies, is distinct from the main objective. Your observation is astute, and we will clarify this distinction in our revised manuscript.
> ### Q7:
> **Query:** "Line 160... Is (3) referencing equation 3?"
>
> **Response:** Indeed, in our manuscript, indicators within parentheses, such as (3), denote references to specific equations. In this context, we aimed to highlight that Equation 3 plays a pivotal role in enabling us to retrieve the expert policy's occupancy measure, which is integral to our algorithm.
> ### Q8:
> **Query:** "What happens for larger alpha" and "It is peculiar in Fig 3 a and b that the over and underestimate already converge for alpha=.5"
>
> **Response:** Our focus was primarily on challenging tasks, with \( $\alpha = 0.1$ \) presenting the most arduous scenario, where merely 10% of the data originates from expert sources. Consequently, we restricted our experiments to \($ \alpha$ \) values within (0, 0.5]. For larger \( $\alpha$ \) values, our algorithm can still emphasize expert data imitation. In scenarios with unknown \( $\alpha$ \), the estimation of \( $\alpha$ \) becomes more precise since the target proportion (pertaining to the expert) exceeds the non-expert proportion. Regarding Fig 3 a and b, the over and underestimation methods converge at \( $\alpha = 0.5$ \) due to their estimation values aligning. Through Equation 7, we deduced that \( $\beta_1 = \frac{1-\alpha}{1+\alpha}$ \) and \( $\beta_2 = \frac{\alpha}{2-\alpha}$ \). When \( $\alpha = 0.5$ \), \( $\beta_1 = \beta_2 = \frac{1}{3}$ \), leading both estimation methods to converge at this point.
>
> ## References
> [1] Confidence-Aware Imitation Learning from Demonstrations with Varying Optimality. NeurIPS 2021.
>
> [2] Learning to Weight Imperfect Demonstrations. ICML 2021.
>
> [3] Imitation Learning from Imperfect Demonstration. ICML 2019.

---

### Author Rebuttal · Authors · 2023-08-10

We are profoundly grateful to all the reviewers for their constructive comments and suggestions. We will address each technical concern or misunderstanding raised by individual reviewers in their respective rebuttal sections. In this general response, we would like to highlight the following updates:

- **PDF file**: An updated version of the HalCheetah and Hopper results has been uploaded to address the queries from Reviewers DWbY and CNdS. We have also included t-tests between COMPILER and COMPILER-E to further address concerns from Reviewer CNdS.

- **Codes and Demonstrations**: The codes for our work are available at this anonymized link: [https://anonymous.4open.science/r/COMPILER-7B6A](https://anonymous.4open.science/r/COMPILER-7B6A). Additionally, the demonstrations used in our experiments can be accessed at: [https://drive.google.com/file/d/1-LWep6U_FHdEWuEJxDaluQXBQWGC086L/view?usp=sharing](https://drive.google.com/file/d/1-LWep6U_FHdEWuEJxDaluQXBQWGC086L/view?usp=sharing).

- **Discussion on Limitations**: We have discussed the limitations of our work in lines 135-136, which address the omission of noise during data collection. Experiments involving noisy $\alpha$ have been included in the Appendix. Furthermore, as mentioned in lines 343-344, we considered a single annotator in VPIL problems. However, real-world scenarios might involve multiple annotators with varying preferences. Based on the reviewers' feedback, we will incorporate a dedicated section to discuss our work's limitations more comprehensively.

---

### Decision · Program_Chairs · 2023-09-21

**Decision:**

Accept (poster)

**Comment:**

The paper proposes an approach that allows imitation learning from mixed expert and non-expert demonstrations. The method allows to disentangle the dataset both when the percentage is known and when it needs to be estimated. The approach is studied with extensive experiments and comparisons against the sota.

The reviewers found the method innovative and sound. The theoretical and experimental analysis are nicely done. The main concerns were about missing details, clarifying the contributions and limitations, inconsistencies/unexplained effects in the results, and some unclear theoretical implications.

The rebuttal managed to overcome all (major) objections of the reviewers. Some reviewers did not react to the rebuttal, but they were already positive about the paper and their concerns have been nicely addressed. Providing the anonymized code was highly appreciated.

All reviewers are now in favor accepting the paper.